# Discovering Latent Network Topology in Contextualized Representations with Randomized Dynamic Programming

## Abstract

The discovery of large-scale discrete latent structures is crucial for understanding the fundamental generative processes of language. In this work, we use structured latent variables to study the representation space of contextualized embeddings and gain insight into the hidden topology of pretrained language models. However, existing methods are severely limited by issues of scalability and efficiency as working with large combinatorial spaces requires expensive memory consumption. We address this challenge by proposing a *Randomized Dynamic Programming* (RDP) algorithm for the approximate inference of structured models with DP-style exact computation (e.g., Forward-Backward). Our technique *samples a subset of DP paths* reducing memory complexity to as small as one percent. We use RDP to analyze the representation space of pretrained language models, discovering a large-scale latent network in a fully unsupervised way. The induced latent states not only serve as anchors marking the topology of the space (neighbors and connectivity), but also reveal linguistic properties related to syntax, morphology, and semantics. We also show that traversing this latent network yields unsupervised paraphrase generation.

## 1 Introduction

The discovery of large-scale discrete latent structures is crucial for understanding the fundamental generative processes of language, and has been shown useful to various NLP tasks ranging from data-to-text generation (Li & Rush, 2020), summarization (Angelidis et al., 2021), syntactic parsing (Kim et al., 2019), and knowledge graph reasoning (Qu et al., 2020). In this work, we use latent structures to analyze geometric properties of representation space of pretrained language models (PLMs). Despite the large volume of recent work analyzing PLMs and proposing various improvements (Rogers et al., 2020), little is known about the topological structure of their representation manifold. Since such structure cannot be easily observed, it is only natural to resort to latent variables. Yet scaling discrete combinatorial structures is extremely difficult with multiple modeling and computational challenges (Wainwright & Jordan, 2008).

In this work, we address the computational challenges arising from working with combinatorial structures. We consider linear-chain CRFs, a popular structured model family (Ma & Hovy, 2016; Sutton & McCallum, 2006) that uses dynamic programming for exact inference. Specifically, we focus on the forward algorithm (Rabiner, 1989), which is widely used to compute the partition function. Space complexity for this algorithm is $O(TN^2)$ where $N$ is the number of latent states and $T$ the length of the sequence. It is precisely the $N^2$ term that becomes problematic when we construct the adjacent gradient graph with automatic differentiation. DP-based inference algorithms are not optimized for modern computational devices like GPUs and typically work under small-data regimes, with $N$ in the range [10, 100] (Ma & Hovy, 2016; Wiseman et al., 2018). With larger $N$, inference becomes intractable since gradients do not easily fit into GPU memory (Sun et al., 2019).

Our algorithmic contribution is a *randomization* technique for dynamic programming which allows us to scale $N$ to thousands (possibly more) latent states. Specifically, to approximate the partition function, instead of summing over *all possible* combinations of latent states, we only sum over paths with *most probable* states, and sample a subset of less likely paths to correct the bias according

to a reasonable proposal. Since we only calculate the sampled path, memory consumption can be reduced to a small controllable budget which is scale invariant. With a larger memory budget, our method becomes more accurate, and our estimation error smaller. We thus recast the memory complexity challenge into a tradeoff between memory budget, proposal accuracy, and estimation error. When applied to linear-chain CRFs, we show that RDP scales the model by *two orders of magnitude* with memory complexity *as small as one percent* of the full DP. Beyond linear-chains, RDP is applicable to any structured model with DP-style exact inference such as trees (Kim et al., 2019) and semi-Markov models (Li & Rush, 2020), and could also be extended to more general message passing algorithms (Wainwright & Jordan, 2008).

Our analytical contribution is a geometric study of the representation manifold of PLMs, using the proposed RDP algorithm. We hypothesize that there exist latent anchor embeddings (or landmarks) that describe the manifold topology. We also expect these anchor states to be informative enough to generate sentences, and their connections to be linguistically meaningful. We induce latent structures using a VAE with an inference model parameterized by a scaled CRF where state-word relations are modeled by the emission potential and state-state transitions are modeled by the transition matrix. The connections of words and states together form a latent network. We use the vector product between contextualized embeddings and state embeddings to parameterize the CRF potentials, *bringing together the geometry of the representation space with graphical model inference*. We further show that it is possible to generate paraphrases by *traversing the induced network*.

Our approach is *fully unsupervised* and the discovered latent network is *intrinsic* to the representation manifold, rather than *imposed* by external supervision, eschewing the criticism of much previous work on supervised probes (Hewitt & Liang, 2019; Chen et al., 2021). In experiments, we first verify the basic properties of RDP (bias-variance) and show its effectiveness for training latent variable models. We then visualize the discovered network based on BERT Devlin et al. (2019), demonstrating how states encode information pertaining to syntax, morphology, and semantics. Finally, we perform unsupervised paraphrase generation by latent network traversal.

## 2 RANDOMIZED DYNAMIC PROGRAMMING

**Preliminaries: Speeding Summation by Randomization**     To motivate our randomized DP, we start with a simple setting, namely estimating the sum of a sorted list. Given a sorted list of positive numbers $\boldsymbol{a}$, naive summation $S = a_1 + ..., +a_N$ requires $N - 1$ addition operations, which is expensive when $N$ is large. Suppose we wish to reduce the number of addition operations to $K_1 << N$, and we already know that the list is long-tailed (similar to how words in language follow a Zipfian distribution such that there are few very high-frequency words that account for most of the tokens in text and many low-frequency words). Then, we only need to sum over the top $K_1$ values to get an efficient estimate:

$$\hat{S}_1 = a_1 + ... + a_{K_1} \qquad \text{where } \{a_i\}_{i=1}^N \text{ sorted, large to small} \tag{1}$$

Clearly, $\hat{S}_1$ underestimates $S$. When the summands are "dense", i.e., not very different from each other, the bias is large because the top $K_1$ terms do not contribute much to the sum (Fig. 1A). To correct this bias, we add samples $a_{\delta_1}, ..., a_{\delta_{K_2}}$ from the remaining summands whose indices $\delta_i$ are sampled from proposal $\delta_i \sim \boldsymbol{q} = [q_{K_1+1}, ..., q_N]$:

$$\hat{S}_2 = a_1 + ... + a_{K_1} + \frac{1}{K_2}\left(\frac{1}{q_{\delta_1}}a_{\delta_1} + ... + \frac{1}{q_{\delta_{K_2}}}a_{\delta_{K_2}}\right) \qquad \delta_i \in \{K_1 + 1, ..., N\} \tag{2}$$

where $K_1 + K_2 = K$. Note that this is an unbiased estimator as $\mathbb{E}[\hat{S}_2] = S$, irrespective of how we choose $\boldsymbol{q}$. Without any knowledge about $\boldsymbol{a}$, the simplest proposal would be uniform, no matter what variance it induces. The more $q_i$ correlates with $a_i$, the less variance $\hat{S}_2$ has. The oracle $q_i$ is proportional to $a_i$, under which $\hat{S}_2$ becomes exact $\hat{S}_2 \equiv S$ as $q_{\delta_i} = a_{\delta_i}/(a_{K+1} + ... + a_N)$ for all $i$. So, the strategy is to exploit our *knowledge* about $\boldsymbol{a}$ to construct a *correlated proposal* $\boldsymbol{q}$. Given this estimator, we can also adjust the computation budget in order to reduce variance. When the distribution is long-tailed, we may increase $K_1$ as an instance of Rao-Blackwellization (Liu et al., 2019). When the distribution is not long-tailed (enough), and top $K_1$ summation underestimates significantly, we may increase $K_2$ to reduce variance, provided we have a fairly accurate $\boldsymbol{q}$, as an instance of importance sampling. This procedure is also discussed in Kool et al. (2020) for

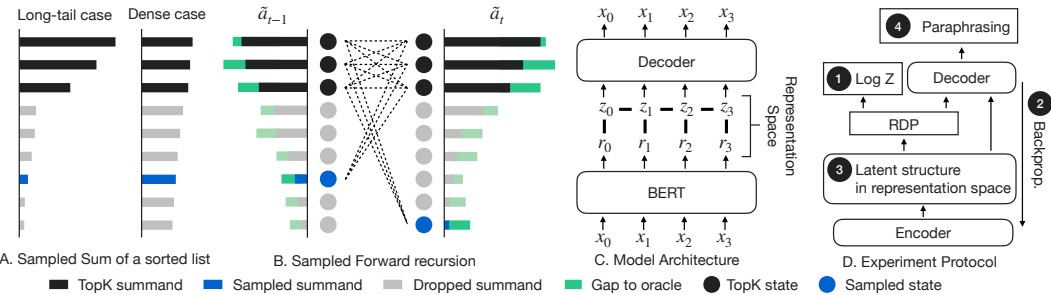

Figure 1: (A): Sampled summation of an array; in the dense case the proposal is important for variance reduction, while in the long-tailed case, topK summands are important; (B): core recursion step of the Randomized Forward algorithm. We get topK and sample from the proposal (black and grey bars); Errors stem from the difference (green bars) between the oracle proposal $\tilde{a}$ and constructed proposal $\tilde{q}$; (C): Inferring latent states within the BERT representation space. We parametrize the CRF factors with vector products; the relations between states and contextualized embeddings together form a latent network (Fig. 3 and 4); (D): Experimental protocol; we first study the basic properties of RDP (steps 1, 2) and then integrate RDP into a LVM for inferring the structure of the representation space (steps 3, 4). Best viewed in color.

gradient estimation. In fact, it is the underlying basis of many Monte Carlo estimators in various settings (Mohamed et al., 2020).

**The Sampled Forward Algorithm**    Now we will show how estimator $\hat{S}_2$ can be used to scale summation in DP. Consider a linear chain CRF which defines a discrete state sequence $\boldsymbol{z} = [z_1, ..., z_T], z_t \in \{1, ..., N\}$ over an input sentence $\boldsymbol{x} = [x_1, ..., x_T]$. Later we will use this CRF to construct an inference model to discover latent network structures within contextualized representations. We are interested in the partition function $Z$ which is commonly computed with the Forward algorithm, a dynamic programming algorithm that sums over the potentials of all possible state sequences. The core recursion steps are:

$$\alpha_{t+1}(i) = \sum_{j=1}^{N} \tilde{a}_{t+1}(i,j) = \sum_{j=1}^{N} \alpha_t(j)\Phi(j,i)\phi(x_{t+1},i) \qquad Z = \sum_{j=1}^{N} \alpha_T(j) \qquad (3)$$

where $\alpha_t(i)$ is the sum of all possible sequences up to step $t$ and at state $i$, $\Phi(\cdot, \cdot)$ is an $N \times N$ transition matrix, and $\phi(x_t, i)$ is the emission potential that models how word $x_t$ generates state $i$. We assume all potentials are positive for simplicity. When implemented on GPUs, space complexity is $O(TN^2)$ (see number of edges in the DP graph in Figure 1B) and it is the squared term $N^2$ that causes memory overflows under automatic differentiation (see Appendix B for engineering details).

Our key insight is to recursively use the memory-efficient randomization of Eq. 2 to estimate Eq. 3 at every step. Given a proposal $\tilde{q}_t$ for each step $t$ that correlates with summands $\tilde{a}_t$ (we discuss how to construct $\tilde{q}_t$ in the next section), we obtain its top $K_1$ index and sample $K_2$ from the rest:

$$[\sigma_{t,1}, ..., \sigma_{t,K_1}, ..., \sigma_{t,N}] = \arg \mathrm{sort}_i \{\tilde{q}_t(i)\}_{i=1}^{N} \qquad (4)$$

$$[\delta_{t,1}, ..., \delta_{t,K_2}] \sim \mathrm{Categorical}\{\tilde{q}_t(\sigma_{t,K_1+1}), ..., \tilde{q}_t(\sigma_{t,N})\} \qquad (5)$$

where $\tilde{q}_t(\cdot)$ are normalized to construct the categorical. Compared to Eq. 3, the key recursion of our Sampled Forward uses the top $K_1$ index $\sigma_t$ and sampled $K_2$ index $\delta_t$ to substitute the full index:

$$\hat{\alpha}_{t+1}(i) = \sum_{j=1}^{K_1} \hat{\alpha}_t(\sigma_{t,j})\Phi(\sigma_{t,j},i)\phi(x_{t+1},i) + \frac{1}{K_2}\sum_{j=1}^{K_2} \frac{\tilde{Z}_t}{\tilde{q}_t(\delta_{t,j})}\hat{\alpha}_t(\delta_{t,j})\Phi(\delta_{t,j},i)\phi(x_{t+1},i) \quad (6)$$

$$\tilde{Z}_t = \sum_{j=K_1+1}^{N} \tilde{q}_t(\sigma_{t,j}) \qquad \hat{Z} = \sum_{j=1}^{K_1} \hat{\alpha}_T(\sigma_{T,j}) + \frac{1}{K_2}\sum_{j=1}^{K_2} \frac{\tilde{Z}_T}{\tilde{q}_T(\delta_{T,j})}\hat{\alpha}_T(\delta_{T,j}) \qquad (7)$$

where the oracle proposal $q_t^*$ is proportional to the actual summand $\tilde{a}_t$ (Eq. 3, a little bit algebra will show this is actually the backward sampling probability $p(z_t = i|z_{t+1} = j)$) , which is only accessible with the full Forward. So, we use the proposal weight $\tilde{q}_t$ (Eq. 4) to move the computation outside the DP. In Fig. 1B, the top $K_1$ summed terms correspond to black nodes. The proposal $\tilde{q}_t$

corresponds to black and grey bars, and its distance from the oracle proposal $\tilde{a}_t$ (which is the major source of variance) is highlighted in green. Sampled indices are shown as blue nodes. Essentially, our Sampled Forward algorithm restricts the DP computation from the full graph to subgraphs with top and sampled edges, reducing complexity to $O(TK^2)$ where $K = K_1 + K_2$. By varying $K$, memory complexity becomes a tradeoff between memory budget and estimation error. By induction, we can show that $\hat{Z}$ (Eq. 7) is an unbiased estimator of $Z$ since $\forall t, \mathbb{E}[\hat{\alpha}_t] = \alpha_t$. When implemented in log space, the expected $\log \hat{Z}$ is a lower bound of the exact $\log Z$ due to Jensen's inequality, and the variance is (trivially) reduced by $\log(\cdot)$. See Appendix for details on implementation (Section C), theoretical analysis (Section A), and extensions to general sum-product structures (Section D).

## 3 LATENT NETWORK TOPOLOGY IN PRETRAINED LANGUAGE MODELS

**Latent States within Representation Space**    We now use the above technique to uncover hidden geometric structures in contextualized representations. In experiments we work with BERT (Devlin et al., 2019) and GPT2 (Radford et al., 2019), however, our method can be easily applied to other pretrained language models. Given sentence $\boldsymbol{x} = [x_1, ..., x_T]$, we denote its contextualized representations as $[\boldsymbol{r}_1, ..., \boldsymbol{r}_T] = \text{PLM}(\boldsymbol{x})$. Representations $\boldsymbol{r}$ for all sentences lie in one manifold $\mathcal{M}$, namely, the representation space of the language model. We hypothesize there exists a set of latent states $\boldsymbol{s}_1, ..., \boldsymbol{s}_M$ that function as *anchors* and outline the space topology. We emphasize that all parameters of the PLM are fixed (i.e., no fine-tuning takes place), so all learned states are *intrinsic* to $\mathcal{M}$. We focus on two topological relations: (a) *state-word* relations, which represent how word embeddings may be summarized by their states and how states can be explained by their corresponding words; and (b) *state-state* relations, which capture how states interact with each other and how their transitions denote meaningful word combinations. Taken together, these two relations form a latent network within $\mathcal{M}$ (visualized in Fig. 3 and 4).

We adopt a minimal parametrization of the inference network so as to respect the intrinsic structure of the representation manifold without imposing strong assumptions (e.g., via regularization). Specifically, for state-word relations, we associate each word embedding $\boldsymbol{r}_t$ with a latent state indexed by $z_t \in \{1, ..., N\}$ (the corresponding embedding of $z_t$ is $\boldsymbol{s}_{z_t}$). For state-state relations, we assume a transition weight $\Phi(i, j)$. Together we have a linear-chain CRF:

$$\log \phi(x_t, z_t) = \boldsymbol{r}_t^\mathsf{T} \boldsymbol{s}_{z_t} \qquad \log \Phi(z_{t-1}, z_t) = \boldsymbol{s}_{z_{t-1}}^\mathsf{T} \boldsymbol{s}_{z_t} \qquad (8)$$

where the dot product follows the common practice of fine-tuning contextualized representations. We use log space for numerical stability. The probability of a state sequence given a sentence is:

$$q_\psi(\boldsymbol{z}|\boldsymbol{x}) = \prod_{t=1}^{T} \Phi(z_{t-1}, z_t)\phi(x_t, z_t)/Z \qquad (9)$$

Here, the only learnable parameters are state embeddings: $\psi = [\boldsymbol{s}_1, ..., \boldsymbol{s}_N]$ as we try to be faithful to the representation manifold. Note how this parametrization reconciles space geometry with graphical models. As $N$ is large, we estimate $Z$ with the proposed Sampled Forward (Eq. 7).

**Constructing the Proposal**    We now return to proposal $\tilde{q}_t$ (Eq. 4) which we construct based on a common observation that linguistic phenomena are long-tailed:

$$\tilde{q}_t(i) \propto \Phi(i)\phi(x_t, i) \qquad \Phi(i) = ||\boldsymbol{s}_i||_1 \qquad (10)$$

where $\phi(x_t, i)$ states that only a few states are likely to generate observation $x_t$, which is often the case in NLP (e.g., there are only a few possible POS tags for each word); and $\Phi(i)$ models the prior probability of state $i$. This choice stems from the empirical observation that larger L1 norm correlates with larger dot product, and is thus more likely to be inferred. Essentially, our proposal combines local emissions $\phi$ and global prior $\Phi$ to approximate the $\tilde{a}_t$ variables (Eq. 3) and bypass their expensive computation.

**Inference and Learning**    We use amortized variational inference to learn $\boldsymbol{s}$. We simply reuse the architecture from previous work Fu et al. (2020); Li & Rush (2020) and build a generative model:

$$p_\theta(\boldsymbol{x}, \boldsymbol{z}) = \prod_t p(x_t|z_{1:t}, x_{1:t-1}) \cdot p(z_t|z_{1:t-1}, x_{1:t-1}) \qquad \boldsymbol{h}_t = \text{Dec}([\boldsymbol{s}_{z_{t-1}}; x_{t-1}], \boldsymbol{h}_{t-1}) \quad (11)$$

$$p(x_t|z_{1:t}, x_{1:t-1}) = \text{softmax}(\text{FF}(\boldsymbol{h}_t)) \qquad p(z_t|z_{1:t-1}, x_{1:t-1}) = \text{softmax}(\text{FF}([\boldsymbol{s}_{z_t}; \boldsymbol{h}_t])) \quad (12)$$

where $\theta$ denotes the decoder parameters, Dec($\cdot$) denotes the decoder (we use an LSTM), $\boldsymbol{h}_t$ denotes decoder states, and FF($\cdot$) denotes a feed-forward network. This autoregressive formulation essentially encourages states to be "generative", i.e., to generate sentences and themselves. We will show in experiments how this formulation lends itself to paraphrasing. We use $q_\psi$ directly from Eq. 9 as our variational posterior, and optimize the following $\beta$-ELBO objective:

$$\mathcal{L}_{\text{ELBO}} = \mathbb{E}_{q_\psi(z|x)}[\log p_\theta(x,z)] - \beta \mathcal{H}(q_\psi(z|x)) \tag{13}$$

where the $\beta$ parameter modulates the topology of the latent structure and prevents posterior collapse. We follow Fu et al. (2020) and use their Gumbel reparameterization to optimize $q_\psi$, which is more stable than the REINFORCE gradient estimator (Li & Rush, 2020).

When integrating RDP with the Gumbel reparameterization, we noticed that the gradient will only pass through the top $K_1$ and sampled $K_2$ states, in other words, not all states receive gradients. In this case, trading $K_1$ against $K_2$ amounts to *exploration* versus *exploitation*. A large $K_1$ means we give gradients to high-confidence states, i.e., we exploit large local emission and global transition potentials. While increasing $K_2$ means we explore low-confidence states. So, by splitting the computation budget between top $K_1$ and sampled $K_2$ states, we not only reduce variance for estimating the partition, but also effectively introduce different strategies for searching over the latent space.

## 4 RELATED WORK

**Efficient Inference for Structured Latent Variables**     There has been substantial interest recently in the application of deep latent variable models (LVMs) to various language related tasks (Wiseman et al., 2018; Li & Rush, 2020), which has also exposed scalability limitations. Earlier attempts to render CRF models efficient (Sokolovska et al., 2010; Lavergne et al., 2010) either make many stringent assumptions (e.g., sparsity), rely on handcrafted heuristics for bias correction (Jeong et al., 2009), or cannot be easily adapted to modern GPUs with tensorization and parallelization. Sun et al. (2019) are closest to our work, however they only consider top $K$ summation and consistently underestimate the partition. Chiu & Rush (2020) scale HMMs but assume words are clustered beforehand. Our approach systematically trades computation with proposal accuracy and estimation error (rather than over-compromising for efficiency). Moreover, we do not impose any hard restrictions like sparsity (Correia et al., 2020), and can accommodate dense and long-tailed distributions. Our method is inspired by randomized automatic differentiation (RAD, Oktay et al., 2020), and can be viewed as RAD applied to the DP computation graph. Advantageously, our proposal is compatible with existing efficient implementations (like Rush, 2020) since it does not change the computation graph.

**Interpretability of Contextualized Representations**     There has been a good deal of interest recently in analyzing contextualized representations and the information they encode. This line of research, collectively known as "Bertology" (Rogers et al., 2020; Hewitt & Manning, 2019), focuses mainly on *supervised* probing of linguistic properties (Tenney et al., 2019), while the geometric properties of the representations have been less studied (Cai et al., 2021). A major dilemma facing this work is whether supervised linguistic probes reveal properties intrinsic to the embeddings or imposed by the supervision signal itself (Hewitt & Liang, 2019; Hall Maudslay et al., 2020; Chen et al., 2021). In this work, we do not use any supervision to ensure that the discovered network is intrinsic to the representation space.

## 5 EXPERIMENTS

In this section, we present our experimental results aimed at analyzing RDP and showcasing its practical utility (see Fig. 1). Specifically, we (1) verify the basic properties of RDP by estimating the partition function and (2) using it to train the structured latent variable model introduced in Section 3; (3) we then turn our attention to pretrained language models and examine the network induced with our approach and whether it is meaningful; and (4) we generate sentence paraphrases by traversing this network. For experiments (1, 2, 4), we use (a). pretrained GPT2 as the encoder since they are more about autoregressive language modeling and generation; (b). the MSCOCO dataset, a common benchmark for paraphrasing (Fu et al., 2019). For experiment (2), we use (a). BERT since it has been the main focus of most previous analytical work (Rogers et al., 2020); (b). the 20News dataset, a popular benchmark for training latent variable models (Grisel et al.). Across all experiments, we

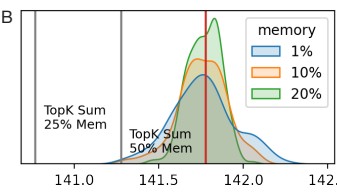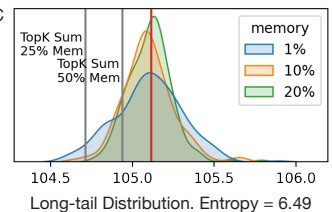

Figure 2: Sampled Forward vs. TopK summation (Sun et al., 2019) in different unit cases during training. Red line: target log partition. Grey line: estimates from TopK. Our method effectively corrects the bias in TopK summation with significantly less memory, and is consistent with dense and long-tailed distributions.

| Model-#States | Dev NLL | Dev PPL | Test NLL | Test PPL |
|---|---|---|---|---|
| FULL-100 (Fu et al., 2020) | 39.64±0.06 | 22.07±0.12 | 39.71±0.07 | 22.32±0.12 |
| TOPK-100 (Sun et al., 2019) | 39.72±0.13 | 22.22±0.23 | 39.76±0.11 | 22.41±0.20 |
| RDP-100 (ours) | **39.59±0.10** | **21.99±0.18** | **39.59±0.08** | **22.12±0.13** |
| TOPK-2K (Sun et al., 2019) | 39.81±0.30 | 22.43±0.44 | 39.84±0.31 | 22.52±0.59 |
| RDP-2K (ours) | **39.47±0.11** | **21.94±0.46** | **39.48±0.14** | **21.93±0.24** |

Table 1: Results on training LVMs on MSCOCO dataset. Models are run 6 times with different random seeds.

use an LSTM decoder with states identical to the encoder (762 for BERT base and GPT2 as in Wolf et al., 2020). More details on experiments and model settings can be found in Appendix E.

## 5.1 BASIC PROPERTIES

We examine the estimation of the partition function for three unit cases, namely dense, intermediate, and long-tailed distributions. Instead of simulating these unit cases, to make our experiments more realistic, we extract CRFs on-the-fly from different LVM training stages. We also study the effects of memory budget by setting $K$ to 20, 200, and 400 (corresponding to 1, 10, and 20 percent of the full memory). We use TopK summation (Sun et al., 2019) as our main baseline. This method can be viewed as setting $K_1 = K$ and $K_2 = 0$ in our framework, i.e., it does not use the random sample. For training LVMs, We consider 100 and 2,000 latent states. With 100 states we are able to perform the summation exhaustively which is the same as Fu et al. (2020). Full summation with 2,000 states is intractable, so we only compare with TopK summation and use $K = 100$.

**Estimating the Partition Function** As shown in Figure 2, TopK summation always underestimates the partition. The gap is quite large in the dense case (large entropy), which happens at the initial stages of training when the model is not confident enough. The long-tailed case represents later training epochs when the model has converged and is more concentrated. Our method effectively corrects the bias, and works well in all unit cases with significantly less memory.

**Training Latent Variable Models** We compare different LVMs in Table 1. Following common practice, we report negative log likelihood (NLL) and perplexity (PPL). We perform an extensive search over multiple hyperparameters (e.g., $\beta$, learning rate, word dropout) across multiple random seeds (3–6) and report the average performance of the best configuration for each method. Our model performs best in both 100 and 2,000 state settings. The advantage is modest (as there are no architecture changes, only different training methods) but consistent. RDP trades off exploitation (i.e., increasing $K_1$) and exploration (i.e., increasing $K_2$) while TopK summation always focuses on the local solutions by passing gradients through top states. Intuitively, we have the chance of discovering better latent states (i.e., larger likelihood) by randomly searching the unexplored space.

## 5.2 DISCOVERING LATENT NETWORKS FROM PRETRAINED EMBEDDINGS

We now discuss how latent structures induced with RDP reveal linguistic properties of contextualized representations. We focus on BERT Devlin et al. (2019) and set the number of latent states to

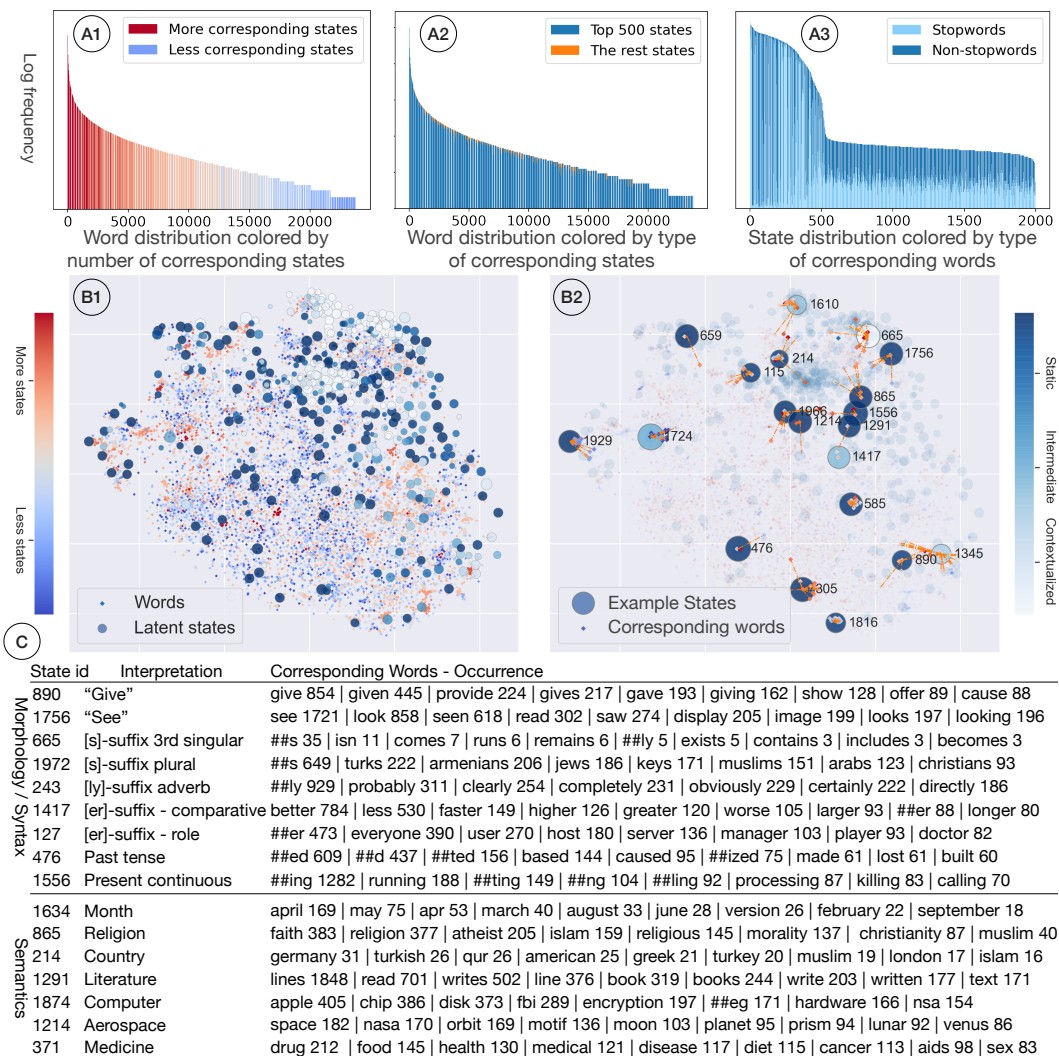

Figure 3: (A1): Frequent words partake in more latent states than rare words (presumably because they are polysemous); (A2 and A3): The distribution of states is also Zipfian, as most frequent states generate most words (the orange portion in A2 is almost indistinguishable); (B): t-SNE (Van der Maaten & Hinton, 2008) visualization of latent network induced from BERT; (B1): Words and their corresponding latent states. For states, the size of circle indicates frequency ($\approx$ aggregated posterior probability) and color thickness means level of contextualization; a state with deeper blue color tends to generate content words (whose meaning is less dependent on context); lighter blue corresponds to stopwords (which are more contextualized); words are also colored by number of states ($\approx$ number of linguistic roles); red color densities mean a word is generated by several states; (B2) and (C): sample from $p^*(x)q_\phi(z|x)$. Our method discovers a spectrum of meaningful states which exhibit both morpholigical, syntactic and semantic functionalities.

2,000. As BERT's vocabulary size is 32K, one state would approximately handle 15 words in the uniform case, functioning as a type of "meta" word. After convergence, we use $q_\psi$ to sample $z$ for each $x$ in the training set (recall we use the 20News dataset). These $z$ can be viewed as samples from the aggregated posterior $\sum_x q_\psi(z|x)p^\star(x)$ where $p^\star(x)$ denotes the empirical data distribution. To get a descriptive summary of BERT's latent topology, we compute the following statistics on $z$ samples: state frequency (Fig. 3, A3); number words corresponding to each state (Fig. 3, A2); number of states corresponding to each word (Fig. 3, A1); and state bigrams (Fig. 4). We further differentiate stopwords (e.g., *in, of, am, is*) from content words.

**State-word Relations** Figure 3 gives a first impression of how latent states spread over the representation space. Overall, we observe that the joint space is Zipfian, and this property characterizes

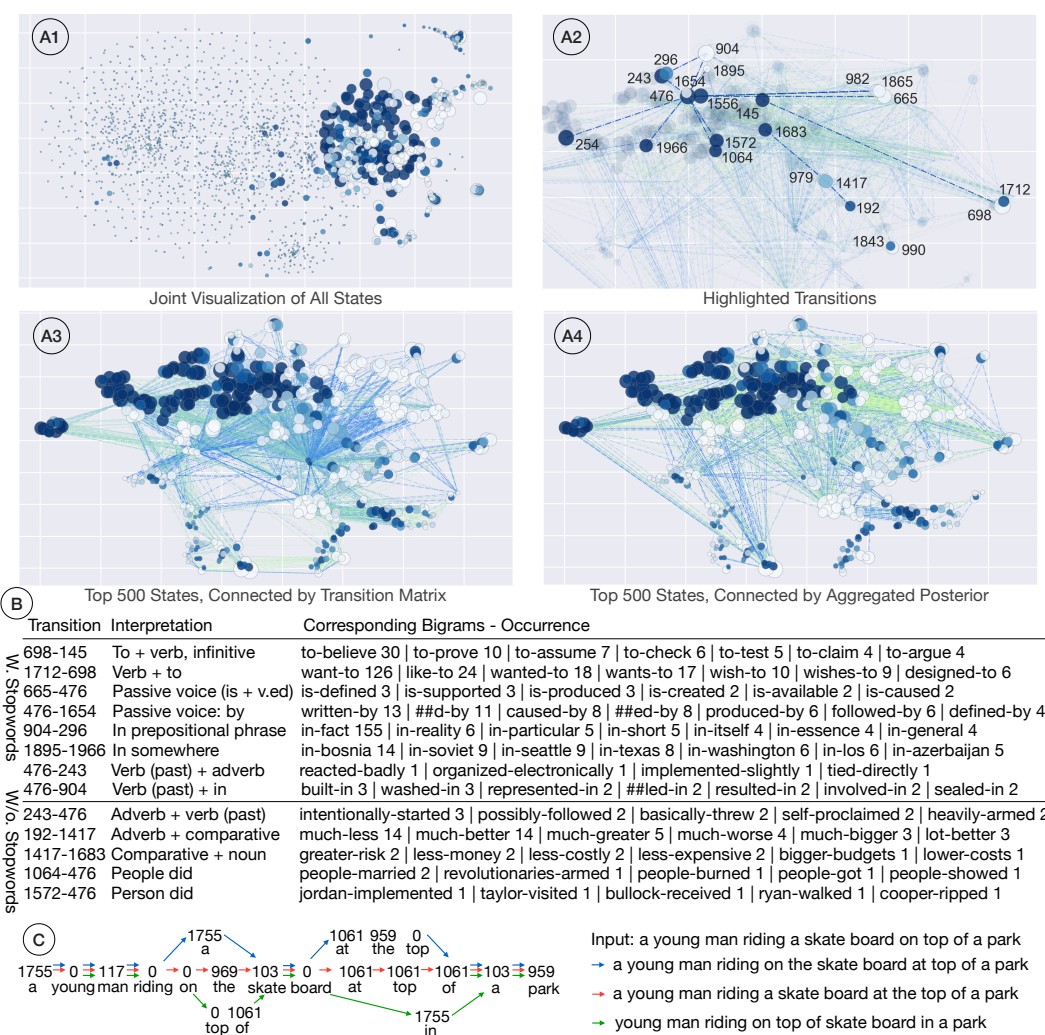

Figure 4: (A1): Geometrical differences between top and tail states; most lexical variations are encoded by the top 500 states while remaining states represent the long tail; (A3 and A4): Network topology within top 500 states; in (A3) nodes are connected to their top 1 neighbor according to the transition matrix $\Phi$ (as a proxy of the empirical prior) and in (A4) according to the most frequent bigram (as a proxy of the aggregated posterior), note how the two are correlated; (A2 and B): Highlighted bigrams and their linguistic interpretation; transitions with stopwords are more about syntax (e.g., *to* with infinitives or transitive verbs); transitions without stopwords are more about specific meanings. (C): paraphrasing as latent network traversal.

the distribution of words (A1), states (A3), and word occurrence within each state (C). We also see that the top 500 states account for most word occurrence (A2) while the remaining states model tail phenomena (A3). We conjecture this number is related to the intrinsic dimension of the data manifold (see Aghajanyan et al. 2021). The induced states encode multiple linguistic properties (Fig. 3, C). Some states are similar to a *lexicon* entry encoding specific words and their morphological variants; other states exhibit clustering based on *morphological* features (*-s, -er, -ly* suffix). We believe this is closely related to the fact that BERT learns embeddings over subwords. Note that the past tense cluster contains words exhibiting both regular (*-ed* suffix) and irregular morphology (e.g., *lost* and *built*). Finally, we also see that some states are largely semantic, similar to a conventional topic model (e.g., Computer and Medicine clusters). See Appendix E.6 for more state-word examples.

**State-State Relations**  As shown in Fig. 4, we observe a clear geometric difference between top and tail states. Most linguistic constructions seem to be captured by the top 500 states (A1). The connections of top states are visualized in (A2–A4). From a statistical perspective, the similarity of (A3) and (A4) clearly shows how the empirical prior (encoded by the transition matrix $\Phi$) matches

| Model | iB4↑ | B4↑ | sB4↓ |
|---|---|---|---|
| CGMH (Miao et al., 2018) | 7.84 | 11.45 | - |
| UPSA (Liu et al., 2020) | 9.26 | 14.16 | - |
| GUMBEL-CRF (Fu et al., 2020) | **10.20** | **15.75** | - |
| GPTNET-50 FULL | 8.81±0.03 | 13.54±0.43 | **33.78±3.78** |
| GPTNET-50 TOPK (Sun et al., 2019) | 8.88±0.04 | 13.84±0.50 | 35.75±4.26 |
| GPTNET-50 RDP (ours) | **9.14±0.18** | **14.33±0.30** | 37.49±4.22 |
| GPTNET-2K TOPK (Sun et al., 2019) | 8.80±0.18 | **14.26±0.30** | 40.21±1.00 |
| GPTNET-2K RDP (ours) | **9.04±0.34** | 13.49±0.55 | **30.97±5.18** |

Table 2: Paraphrase generation on the MSCOCO dataset (Fu et al., 2019). Numbers in first block taken from Fu et al. (2020). We report model performance using BLEU 4gram (B4), self BLEU 4gram (sB4), and iBLUE (iB4). Performance is averaged over 3 random seeds.

the aggregated posterior (encoded in the bigram sample from $q_\psi$), which is an important desideratum of generative modeling (Mathieu et al., 2019). Note that the number of edges linked to each node, again, follows a Zipfian distribution as top nodes have most of the connections. From a linguistic perspective, we see how the combination of states leads to meaningful syntactic and semantic constructions. Again, BERT encodes various syntactic configurations such as *to* infinitives, passive voice, and even manages to distinguish adverbials (e.g., *in fact*) from prepositional phrases (e.g., *in Bosnia*). In general, the latent network seems to have some grasp of syntax, semantic roles, and collocations. In the following section, we examine whether this inherent knowledge can be harvested for generation. See Appendix E.7 for more state transition examples.

### 5.3 PARAPHRASING THROUGH NETWORK TRAVERSAL

We now study how the latent network can be usefully employed to generate paraphrases without access to parallel training instances. Given a sentence, we generate its paraphrase by conditioning on the input which we represent as a bag-of-words Fu et al. (2020) *and* by sampling from latent states. This amounts to traversing the latent network then fill in the traversal path to assemble a sentence, as visualized in Fig. 4 C. We instantiate our approach with a latent network learned from GPT2 representations (Radford et al., 2019) and refer to our model collectively as GPTNET.

We compare against three previous unsupervised models (first block in Table 2), including CGMH (Miao et al., 2019), a general-purpose MCMC method for controllable generation; UPSA (Liu et al., 2020), a strong paraphrasing model with simulated annealing, and GUMBEL-CRF (Fu et al., 2020), a template induction model based on a continuous relaxation of the CRF sampling algorithm. We present GPTNET variants with 50 and 2,000 states, and show results with RDP and topK, and the full summation for 50 states. Following previous work, we use iBLEU (Sun & Zhou, 2012) as our main metric, which trades off fidelity to the references (BLEU) and variation from the input (self-BLEU). Table 2 shows that RDP is superior to TopK and full summation in terms of iBLUE. GPTNet models do not outperform GUMBEL-CRF or UPSA. This is expected as these methods are highly tailored to the task and more flexible (e.g., they do not fix the encoder), while we restrict the modeling within the GPT2 representation space (to infer its structure). So, our results should be viewed as a sanity check demonstrating the latent network is indeed meaningful for generation (see Appendix E.8 for more generation examples).

## 6 CONCLUSION

In this paper, we have developed a general method for scaling the inference of structured latent variable models with randomized dynamic programming. It is a useful tool for the visualization and inspection of the intrinsic structure of contextualized representations. Experiments with BERT reveal the topological structure of its latent space: state-word connections encapsulate syntactic and semantic roles while state-state connections correspond to phrase constructions. Moreover, traversal over a sequence of states represents underlying sentence structure.

**Ethics Statement**  As this paper inspects the internal structure of pretrained language models, it is likely that it will reveal frequent linguistic patterns encoded in the language model. Specifically, the frequent words, phrases, and sentences associated with different gender, ethnic groups, nationality, interest groups, social status, and all other factors, are likely to be revealed by our model. When calling the generative part of our model for paraphrasing, these differences are likely to exist in the generated sentences (depending on the dataset). These facts should be considered when using this model.

**Reproducibility Statement**  A step-by-step implementation guide for our randomized forward algorithm is provided in Appendix section C. The comparison of RDP versus other possible solutions for scaling the structured models is provided in Appendix section B. A detailed description of the model architecture is provided in the Appendix section E.1. A detailed description of data processing is provided in the Appendix section E.2. A detailed description of training strategy, hyperparameter search strategy, and model selection, is provided in Appendix section E.3. A detailed description of visualization procedure is provided in Appendix section E.5. We will release code after the anonymity period.

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

**Appendix, table of contents**

## A  THEORETICAL ANALYSIS OF SAMPLED FORWARD ALGORITHM

### A.1  BIAS ANALYSIS

In this section, we discuss the unbiasedness and variance of the Randomized Forward algorithm. We first show that Randomized Forward gives an unbiased estimator of the partition function.

**Theorem A.1** (Unbiasedness). *For all $t \in [1, 2, ..., T]$, the sampled sum $\hat{\alpha}_t$ (Eq. 6) is an unbiased estimator of the forward variable $\alpha_t$ (Eq. 3). The final sampled sum $\hat{Z}$ (Eq. 7) is an unbiased estimator of the partition function $Z$ (Eq. 3).*

*Proof.* By the Second Principle of Mathematical Induction. Assume initialization $\alpha_1(i) = \phi(x_1, i)$. Firstly at $t = 2$, for all $i$, we have:

$$\mathbb{E}_{q_1}[\hat{\alpha}_2(i)] = \sum_{j=1}^{K_1} \alpha_1(\sigma_{1,j})\Phi(\sigma_{1,j}, i)\phi(x_2, i) + \frac{1}{K_2} \underbrace{\sum_{j=1}^{K_2} \mathbb{E}_{q_1}\left[\frac{\tilde{Z}_1}{\tilde{q}_1(\delta_{1,j})}\alpha_1(\delta_{1,j})\Phi(\delta_{1,j})\phi(x_2, i)\right]}_{=A} \tag{14}$$

where the second term can be expanded as a masked summation with the index rearranged from $\sigma_{2,K_1+1}$ to $\sigma_{2,N}$:

$$A = \sum_{j=1}^{K_2} \mathbb{E}_{q_1}\left[\frac{\tilde{Z}_1}{\tilde{q}_1(\delta_{1,j})}\alpha_1(\delta_{1,j})\Phi(\delta_{1,j})\phi(x_2, i)\right] \tag{15}$$

$$= \sum_{k=1}^{K_2}\sum_{j=K_1+1}^{N} \mathbb{E}_{q_1}\left[\frac{1}{q_1(\delta_{1,k})}\mathbb{1}(\delta_{1,k} = j)\alpha_1(\sigma_{1,j})\Phi(\sigma_{1,j})\phi(x_2, i)\right] \tag{16}$$

$$= \sum_{k=1}^{K_2}\sum_{j=K_1+1}^{N} \underbrace{\mathbb{E}_{q_1}\left[\frac{1}{q_1(\delta_{1,k})}\mathbb{1}(\delta_{1,k} = j)\right]}_{=1}\alpha_1(\sigma_{1,j})\Phi(\sigma_{1,j})\phi(x_2, i) \tag{17}$$

$$= K_2 \sum_{j=K_1+1}^{N} \alpha_1(\sigma_{1,j})\Phi(\sigma_{1,j})\phi(x_2, i) \tag{18}$$

Notice how we re-index the sum. Now put it back to Eq. 14 to get:

$$\mathbb{E}_{q_1}[\hat{\alpha}_2(i)] = \sum_{j=1}^{K_1} \alpha_1(\sigma_{1,j})\Phi(\sigma_{1,j}, i)\phi(x_2, i) + \frac{1}{K_2} \cdot K_2 \sum_{j=K_1+1}^{N} \alpha_1(\sigma_{1,j})\Phi(\sigma_{1,j})\phi(x_2, i) \tag{19}$$

$$= \sum_{j=1}^{N} \alpha_1(\sigma_{1,j})\Phi(\sigma_{.j}, i)\phi(x_2, i) \tag{20}$$

$$= \sum_{j=1}^{N} \alpha_1(j)\Phi(j, i)\phi(x_2, i) \tag{21}$$

$$= \alpha_2(i) \tag{22}$$

This verifies the induction foundation that $\mathbb{E}_{q_1}[\hat{\alpha}_2(i)] = \alpha_2(i)$ for all $i$.

Now assume for time index $t$ we have $\forall i, \mathbb{E}_{q_{1:t-1}}[\hat{\alpha}_t(i)] = \alpha_2(i)$. Consider $t + 1$, we have:

$$\mathbb{E}_{1:q_t}[\hat{\alpha}_{t+1}(i)] = \sum_{j=1}^{K_1} \mathbb{E}_{q_{1:t-1}}\big[\hat{\alpha}_t(\sigma_{t,j})\big]\Phi(\sigma_{t,j},i)\phi(x_{t+1},i) \tag{23}$$

$$+ \frac{1}{K_2}\sum_{j=1}^{K_2} \mathbb{E}_{q_{1:t-1}}\big[\hat{\alpha}_t(\delta_{t,j})\big] \cdot \mathbb{E}_{q_t}\big[\frac{\tilde{Z}_t}{\tilde{q}_t(\delta_{t,j})}\Phi(\delta_{t,j},i)\phi(x_{t+1},i)]\big] \tag{24}$$

$$= \sum_{j=1}^{K_1} \alpha_t(\sigma_{t,j})\Phi(\sigma_{t,j},i)\phi(x_{t+1},i) \tag{25}$$

$$+ \underbrace{\frac{1}{K_2}\sum_{j=1}^{K_2} \alpha_t(\sigma_{t,j}) \cdot \mathbb{E}_{q_t}\big[\frac{\tilde{Z}_t}{\tilde{q}_t(\delta_{t,j})}\Phi(\delta_{t,j},i)\phi(x_{t+1},i)]\big]}_{=A'} \tag{26}$$

Note how we decompose the expectation by using the independence: $q_{1:t} = q_{1:t-1} \cdot q_t$ With a similar masked summation trick as $A$, we have:

$$A' = K_2 \sum_{j=K_1+1}^{N} \alpha_t(\sigma_{t,j})\Phi(\sigma_{t,j},i)\phi(x_{t+1},i) \tag{27}$$

This gives us:

$$\mathbb{E}_{1:q_t}[\hat{\alpha}_{t+1}(i)] = \sum_{j=1}^{K_1} \alpha_t(\sigma_{t,j})\Phi(\sigma_{t,j},i)\phi(x_{t+1},i) + \frac{1}{K_2} \cdot K_2 \sum_{j=K_1+1}^{N} \alpha_t(\sigma_{t,j})\Phi(\sigma_{t,j},i)\phi(x_{t+1},i) \tag{28}$$

$$= \sum_{j=1}^{N} \alpha_t(\sigma_{t,j})\Phi(\sigma_{t,j},i)\phi(x_{t+1},i) \tag{29}$$

$$= \sum_{j=1}^{N} \alpha_t(j)\Phi(j,i)\phi(x_{t+1},i) \tag{30}$$

$$= \alpha_{t+1}(i) \tag{31}$$

Thus showing $\hat{\alpha}_t$ is an unbiased estimator for $\alpha_t$ at each step $t$. Setting $t = T$, the last step, gives us $\mathbb{E}[\hat{Z}] = Z$ (details similar to the above). $\qquad\square$

**Corollary A.1.1.** *When we change the (sum, product) semiring to the (log-sum-exp, sum) semiring, the expectation of the estimator will become a lower bound of $\log\alpha_t$ and $\log Z$.*

*Proof.* Denote $l_t(i) = \log\alpha_t(i)$ and $\Omega$ the set of sampled indices where $|\Omega| = K_2$. For simplicity, we omit the top $K_1$ summation and only show the summation of the sample. Cases where $t > 2$ can be derived similarly as following:

$$\hat{l}_2(i) = \log\sum_{j\in\Omega}\exp(l_1(j) + \log\Phi(j,i) + \log\phi(x_t,i)) - \log K_2 \tag{32}$$

$$\mathbb{E}[\hat{l}_2(i)] = \mathbb{E}\bigg[\log\sum_{j\in\Omega}\exp(l_1(j) + \log\Phi(j,i) + \log\phi(x_t,i))\bigg] - \log K_2 \tag{33}$$

$$\leq \log\mathbb{E}\bigg[\sum_{j\in\Omega}\exp(l_1(j) + \log\Phi(j,i) + \log\phi(x_t,i))\bigg] - \log K_2 \tag{34}$$

$$= \log K_2 - \log K_2 + \log\sum_{j\in\Omega}\exp(l_1(j) + \log\Phi(j,i) + \log\phi(x_t,i)) \tag{35}$$

$$= l_2(i) \tag{36}$$

where Eq. 34 comes from Jensen's inequality. Then by induction one can show at everystep, we have $\mathbb{E}[\hat{l}_t(i)] \leq l_t(i)$. $\qquad\square$

Although implementation in the log space makes the estimate biased, it reduces the variance exponentially in a rather trivial way. It also provides numerical stability. So, in practice we use it for training.

## A.2 VARIANCE ANALYSIS

Now we analyze variance. We start with the estimator $a_\delta/q_\delta$, $\delta \sim \text{Categorical}\{q_{K_1+1}, ...q_{K_N}\}$ in Eq. 2. Firstly, this is an unbiased estimator of the tail sum:

$$\mathbb{E}[a_\delta/q_\delta] = \sum_{i=K_1+1}^{N} a_i \tag{37}$$

We have the folling variance arguments:

**Theorem A.2** (Variance of the tail estimator).

$$\mathbb{V}[a_\delta/q_\delta] = \sum_{i=K_1+1}^{N} a_i^2/q_i - \left(\sum_{i=K_1+1}^{N} a_i\right)^2 \tag{38}$$

$$= S_{K_1}^2 \left(\sum_{i=K_1+1}^{N} \epsilon_i^2/q_i + 2\epsilon_i\right) \tag{39}$$

*where*

$$S_{K_1} = \sum_{i=K_1+1}^{N} a_i \qquad \epsilon_i = a_i/S_{K_1} - q_i \tag{40}$$

*which says the variance is:*

- *quadratic to the tail sum $S_{K_1}$, which will can be reduced by increasing $K_1$ as the effect of Rao-Blackwellization.*

- *approximately quadratic to the gap $\epsilon_i$, as the differences between the proposal $q_i$ and the oracle $a_i/S_{K_1}$, which can be reduced by choosing a correlated proposal as the effect of importance sampling.*

*Optimal zero variance is achieved on*

$$q_i^* = a_i/S_{K_1} \tag{41}$$

*Proof.* The variance is then:

$$\mathbb{V}[a_\delta/q_\delta] = \mathbb{E}[(a_\delta/q_\delta)^2] - \mathbb{E}[a_\delta/q_\delta]^2 \tag{42}$$

where the first term is the second moment:

$$(a_\delta/q_\delta)^2 = \left(\sum_{i=K_1+1}^{N} a_i \mathbb{1}[\delta = i]/q_i\right)^2 \tag{43}$$

$$= \sum_{i=K_1+1}^{N} a_i^2 \mathbb{1}[\delta = i]/q_i^2 + 2 \underbrace{\sum_{i=K_1+1}^{N} \sum_{j=K_1+1}^{N} \frac{a_i a_j \mathbb{1}[\delta = i]\mathbb{1}[\delta = j]}{q_i q_j}}_{=0} \tag{44}$$

$$= \sum_{i=K_1+1}^{N} a_i^2 \mathbb{1}[\delta = i]/q_i^2 \tag{45}$$

taking the expection of it we get:

$$\mathbb{E}[(a_\delta/q_\delta)^2] = \mathbb{E}[\sum_{i=K_1+1}^{N} a_i^2 \mathbb{1}[\delta=i]/q_i^2] \tag{46}$$

$$= \sum_{i=K_1+1}^{N} a_i^2/q_i \tag{47}$$

Plug this back, variance is:

$$\mathbb{V}[a_\delta/q_\delta] = \sum_{i=K_1+1}^{N} a_i^2/q_i - (\sum_{i=K_1+1}^{N} a_i)^2 \tag{48}$$

To get the optimal proposal, we solve the constrained optimization problem:

$$\min_{q_i} \sum_{i=K_1+1}^{N} a_i^2/q_i - (\sum_{i=K_1+1}^{N} a_i)^2 \tag{49}$$

$$s.t. \sum_{i=K_1+1}^{N} q_i = 1 \tag{50}$$

By solving the corresponding Lagrangian equation system (omitted here), we get the optimal value achieved at:

$$q_i^* = \frac{a_i}{\sum_{i=K_1+1}^{N} a_i} \qquad \sum_{i=K_1+1}^{N} a_i^2/q_i^* - (\sum_{i=K_1+1}^{N} a_i)^2 = 0 \tag{51}$$

This says zero variance is achieved by a proposal equal to the normalized summands.

Then define the gap between the proposal and the normalized summands as:

$$\epsilon_i = \frac{a_i}{S_{K_1}} - q_i \tag{52}$$

Plug this to the variance expression we get:

$$\mathbb{V}[a_\delta/q_\delta] = S_{K_1}^2 (\sum_{i=K_1+1}^{N} \epsilon_i^2/q_i + 2\epsilon_i) \tag{53}$$

$$\square$$

**Corollary A.2.1.** *When increasing the sample size to $K_2$, the variance will reduce to*

$$\mathbb{V}[\frac{1}{K_2} \sum_{j=1}^{N} \frac{a_{\delta_j}}{q_{\delta_j}}] = \frac{1}{K_2} S_{K_1}^2 (\sum_{i=K_1+1}^{N} \epsilon_i^2/q_i + 2\epsilon_i) \tag{54}$$

Now we consider the variance of the Sampled Forward algorithm. An exact computation would give complicated results. For simplification, we give an asymptotic argument with regard to Rao-Blackwellization and importance sampling:

**Theorem A.3** (Single Step Asymptotic Variance of Sampled Forward). *At each step, the alpha varible estimator has the following asymptotic variance:*

$$\mathbb{V}[\hat{\alpha}_t(i)] = O\Big(\frac{1}{K_2} \alpha_{t,K_1}^2(i) \cdot \epsilon_t^2(i)\Big) \tag{55}$$

*where:*

- $\alpha_{t+1,K_1}(i) = \sum_{j=K_1+1}^{N} \tilde{\alpha}_t(j,i) = \sum_{j=K_1+1}^{N} \sqrt{\mathbb{E}_{q_{1:t-1}}[\hat{\alpha}_t^2(j)]} \Phi(j,i)\phi(x_t,i)$ *is a tail sum after the top $K_1$ summands. This term will reduce if we increase $K_1$, as an instance of Rao-Blackwellization.*

- $\epsilon_t^2(i) = \sum_{j=K_1+1}^N \epsilon_{t-1}^2(j,i)/q_{t-1}(j)$ *and* $\epsilon_{t-1}(j,i)$ *is the difference between the proposal* $q_{t-1}(j)$ *and the oracle proposal in Eq. 3. This term will reduce if the proposal is more correlated to the oracle, as an instance of Importance Sampling.*

*Proof.* We start with a simple setting where $K_2 = 1$. At step $t + 1$ we have the following variance recursion:

$$\mathbb{E}_{q_{1:t}}[\hat{\alpha}_{t+1}^2(i)] = \sum_{j=K_1+1}^N \frac{\Phi^2(j,i)\phi^2(x_{t+1},i)}{q_t(j)} \cdot \mathbb{E}_{q_{1:t-1}}[\hat{\alpha}_t^2(j)] \tag{56}$$

This is derived by plugging estimator 6 to the variance Eq. 38 we have just derived. Denote:

$$\alpha_{t+1,K_1}(i) = \sum_{j=K_1+1}^N \tilde{\alpha}_t(j,i) = \sum_{j=K_1+1}^N \sqrt{\mathbb{E}_{q_{1:t-1}}[\hat{\alpha}_t^2(j)]}\Phi(j,i)\phi(x_{t+1},i) \tag{57}$$

Then we have

$$\mathbb{V}_{q_{1:t}}[\hat{\alpha}_{t+1}(i)] = \alpha_{t+1,K_1}^2(i)\Big( \sum_{j=K_1+1}^N \frac{\epsilon_t^2(j,i)}{q_t(i)} + 2\epsilon_t(j,i)\Big) \tag{58}$$

where $\epsilon_t(j,i)$ is the differences between the proposal and the normalized exact summands at step $t$ state $i$:

$$\epsilon_t(j,i) = \frac{\tilde{\alpha}_t(j,i)}{\sum_{j=K_1=1}^N \tilde{\alpha}_t(j,i)} - q_t(j) \tag{59}$$

Dropping out the first order errors and increasing the number of sample to $K_2$, we have the asymptotics:

$$\mathbb{V}[\hat{\alpha}_t(i)] = O\Big(\frac{1}{K_2}\alpha_{t,K_1}^2(i) \cdot \epsilon_t^2(i)\Big) \tag{60}$$

$\square$

**Theorem A.4** (Asymptotic Variance of Sampled Forward Partition Estimation). *The alpha variable estimators has the following asymptotic variance recurrsion:*

$$\mathbb{V}[\hat{\alpha}_{t+1}(i)] = O(\frac{1}{K_2} \cdot \phi_{t,K_1}^2 \cdot \epsilon_{t,K_1}^2 \cdot \mathbb{V}[\hat{\alpha}_t]) \tag{61}$$

*Compared with Eq. 55, this expression:*

- *Uses the product of the factors $\phi_{t,K_1}$ (a function of the sum-prod of the factor at step $t$) and the previous step variance $\mathbb{V}[\hat{\alpha}_t]$ to substitute the $\alpha_{t,K_1}$ in equation 55. Again, this term will decrease with a larger $K_1$ (Rao-Blackwellization).*

- $\epsilon_{t,K_1}^2(i) = \sum_{j=K_1+1}^N \epsilon_{t-1}^2(j,i)/q_{t-1}(j)$ *and* $\epsilon_{t-1}(j,i)$ *is the difference between the proposal $q_{t-1}(j)$ and the oracle proposal in Eq. 3. This term will reduce if the proposal is more correlated to the oracle, as an instance of Importance Sampling (same as Eq. 55).*

*Consequently, the partition function has the following asymptotic variance:*

$$\mathbb{V}[\hat{Z}] = O(\prod_{t=1}^T \frac{1}{K_2} \cdot \phi_{t,K_1}^2 \cdot \epsilon_{t,K_1}^2) \tag{62}$$

*When implemented in the log space, the variance is trivially reduced exponentially:*

$$\mathbb{V}[\log \hat{Z}] = O(\sum_{t=1}^T \log \frac{1}{K_2} + 2\log \phi_{t,K_1} + 2\log \epsilon_{t,K_1}) \tag{63}$$

*Proof.* Firstly for simplcity we assume $K_1 = 0$ and $K_2 = 1$. The the estimator variance is:

$$\mathbb{V}[\hat{\alpha}_{t+1}(i)] = \mathbb{E}_{q_{1:t}}[\hat{\alpha}_{t+1}^2(i)] - \alpha_{t+1}^2(i) \tag{64}$$

$$= \mathbb{E}_{q_{1:t-1}}\left[\sum_{j=1}^N \frac{\hat{\alpha}_t^2(j)\Phi^2(j,i)\phi^2(x_t,i)}{q_t(j)} \cdot \frac{\alpha_t^2(j)}{\alpha_t^2(j)}\right] - \alpha_{t+1}^2(i) \tag{65}$$

Recall:

$$\alpha_{t+1}(i) = \sum_{j=1}^N \alpha_t(j)\phi(j,i)\phi(x_t,i) \tag{66}$$

Let:

$$\epsilon_t(j,i) = \underbrace{\frac{\alpha_t(j)\phi(j,i)\phi(x_t,i)}{\alpha_{t+1}(i)}}_{=q(z_t=j|z_{t+1}=i)} - q_t(j) \tag{67}$$

Then:

$$\mathbb{V}[\hat{\alpha}_{t+1}(i)] = \mathbb{E}_{q_{1:t-1}}\left[\sum_{j=1}^N \alpha_{t+1}^2(i)\left(\frac{\epsilon_t^2(j,i)}{q_t(j)} + 2\epsilon_t(j,i) + q_t(j)\right)\frac{\hat{\alpha}_t^2(j)}{\alpha_t^2(j)}\right] - \alpha_{t+1}^2(i) \tag{68}$$

$$= \alpha_{t+1}^2(i)\left(\sum_{j=1}^N \left(\frac{\mathbb{E}_{q_{1:t-1}}[\hat{\alpha}_t^2(j)]}{\alpha_t^2(j)}q_t(j)\right) + \sum_{j=1}^N \left(\frac{\epsilon_t^2(j,i)}{q_t(j)} + 2\epsilon_t(j,i)\right)\frac{\mathbb{E}_{q_{1:t-1}}[\hat{\alpha}_t^2(j)]}{\alpha_t^2(j)}\right) \tag{69}$$

$$- \alpha_{t+1}^2(i) \tag{70}$$

Note that there exist $J_t$ such that:

$$\frac{\mathbb{E}_{q_{1:t-1}}[\hat{\alpha}_t^2(j)]}{\alpha_t^2(j)} <= J_t \tag{71}$$

This is because of the bounded gap of the Jensen's inequality. Also recall:

$$\sum_{j=1}^N q_t(j) = 1 \tag{72}$$

So we get:

$$\mathbb{V}[\hat{\alpha}_{t+1}(i)] <= \alpha_{t+1}^2(i)\left(J_t - 1 + \sum_{j=1}^N \left(\frac{\epsilon_t^2(j,i)}{q_t(j)} + 2\epsilon_t(j,i)\right) \cdot \left(\frac{\mathbb{V}[\hat{\alpha}_t(j)]}{\alpha_t^2(j)} - 1\right)\right) \tag{73}$$

$$= \alpha_{t+1}^2(i)\left(J_t - 1 - \sum_{j=1}^N \left(\frac{\epsilon_t^2(j,i)}{q_t(j)} + 2\epsilon_t(j,i)\right)\right) \tag{74}$$

$$+ \alpha_{t+1}^2(i)\left(\sum_{j=1}^N \left(\frac{\epsilon_t^2(j,i)}{q_t(j)} + 2\epsilon_t(j,i)\right) \cdot \left(\frac{\mathbb{V}[\hat{\alpha}_t(j)]}{\alpha_t^2(j)}\right)\right) \tag{75}$$

empirically $J_t$ is not the dominate source of variance (but could be in the worst case, depending on the tightness of Jensen's inequality). We focus on the second term:

$$\mathbb{V}[\hat{\alpha}_{t+1}(i)] = O\left(\alpha_{t+1}^2(i)\left(\sum_{j=1}^N \left(\frac{\epsilon_t^2(j,i)}{q_t(j)} + 2\epsilon_t(j,i)\right) \cdot \left(\frac{\mathbb{V}[\hat{\alpha}_t(j)]}{\alpha_t^2(j)}\right)\right)\right) \tag{76}$$

$$= O\left(\left(\sum_{j=1}^N \alpha_t(j)\underbrace{\Phi(j,i)\phi(x_t,j)}_{O(\phi_t^2)}\right)^2 \cdot \left(\underbrace{\sum_{j=1}^N \left(\frac{\epsilon_t^2(j,i)}{q_t(j)} + 2\epsilon_t(j,i)\right) \cdot \frac{1}{\alpha_t^2(j)} \cdot \underbrace{\mathbb{V}[\hat{\alpha}_t(j)]}_{O(\mathbb{V}[\alpha_t])}}_{O(\epsilon_t^2)}\right)\right) \tag{77}$$

$$= O(\phi_t^2 \cdot \epsilon_t^2 \cdot \mathbb{V}[\alpha_t]) \tag{78}$$

Note that a lot of higher-order sum-products are simplified here. Adding top $K_1$ summation and increasing the sample size to $K_2$ leads to variance reduction as:

$$\mathbb{V}[\hat{\alpha}_{t+1}(i)] = O(\frac{1}{K_2} \cdot \phi_{t,K_1}^2 \cdot \epsilon_{t,K_1}^2 \cdot \mathbb{V}[\alpha_t]) \tag{79}$$

Recursively expand this equation we get:

$$\mathbb{V}[\hat{Z}] = O(\prod_{t=1}^{T} \frac{1}{K_2} \cdot \phi_{t,K_1}^2 \cdot \epsilon_{t,K_1}^2) \tag{80}$$

Chaning the implementation to the log space we reduce the variance exponentially:

$$\mathbb{V}[\log \hat{Z}] = O(\sum_{t=1}^{T} \log \frac{1}{K_2} + 2\log \phi_{t,K_1} + 2\log \epsilon_{t,K_1}) \tag{81}$$

$\square$

# B  CHALLENGES OF FULL DP UNDER AUTOMATIC DIFFERENTIATION

Now we analyze in detail why direct full summation causes memory overflow. Firstly, we note that in a bachified stochastic gradient descent setting, memory complexity for the forward algorithm is

$$O(BTN^2) \tag{82}$$

Where $B$ is the batch size, $T$ is the maximum sequence length after padding, and $N$ is the number of states. Consider a typical setting where $B = 100, T = 100, N = 1000$, then the complexity is:

$$\mathcal{C} = c \times 100 \times 100 \times 1000 \times 1000 = c \times 10^{10} \tag{83}$$

where $c$ is a constant depending on specific hardware and libraries. At first sight this may seem reasonable with a large GPU memory (e.g., 16G). Also, one can come up with improvements, such as not storing the intermediate $\alpha_t$ variables. If we only need one single forward pass, such engineering tricks are indeed effective. In our experiments, when setting $B = 50, T = 15$, and $N = 2000$, the actual memory consumption for the forward computation, together with other model components, is about 4G when implemented with PyTorch 1.8.0.

However, this is not the case when working under an automatic differentiation (AD) setting. If we want to compute the gradients of the partition function (e.g., to optimize the likelihood), we inevitably need to store *all the intermediate steps* to keep the full computation graph. Then, the adjacent graph, in principle, has the same complexity as the forward graph. But in practice, it will be more complicated due to the actual implementation of AD engines. Following the above example where $B = 50, T = 15, N = 2000$, we get a memory overflow on a 16G memory GPU when calling the PyTorch backward function. This situation also immediately invalidates many engineering tricks (e.g., we cannot drop the intermediate $\alpha_t$ variables anymore), and substantially increases the difficulty of coming up with new ones, since we would also be working with the internal mechanism of automatic differentiation engines. Note that the internal mechanism of these engines is complicated and opaque to general practitioners (e.g., they may change the underlying computation graph for better speed), and many of these engines (like PyTorch) are not optimized for dynamic programming.

In fact, it would be unwise to overwrite the backward computation, even if the AD engine allowed us to do so, as it would significantly increase engineering difficulty, as we not only need to overwrite the first order gradients (like the gradients for the likelihood), but the second order gradients too (e.g., gradients for marginals or reparameterized samples). In fact, a brute force implementation would not only forego all the advantages of AD engines (like operator-level optimization), but would also require separate implementations for every graph (e.g., chains, trees, semi-Markovs, and Ising-Potts), every inference algorithm (e.g., partition, marginal, sampling, reparameterization, entropy), and higher order gradients. In summary, this is an extremely difficult path that requires clever tricks to efficiently improve a large table of already complicated DP algorithms.

This is why we do not interfere with the AD framework and work on more general and efficient algorithms, rather than tailored implementation tricks. With the randomized DP, *we do not re-implement*

*anything*. We only choose the index by the pre-computed proposal, and introduce the re-indexed potentials to existing efficient libraries (like Torch-struct in Rush, 2020). This is least difficult from an implementation perspective and best exploits the power of efficient structured prediction libraries and AD engines.

## C   IMPLEMENTATION OF SAMPLED FORWARD ALGORITHM

Now we discuss how to implement the sample forward in detail. Our implementation aims to set all computation outside the actual DP, and reuse existing optimized DP libraries. Specifically, we:

1. normalize the log potentials;
2. construct the proposal according to the local potential and a global prior;
3. obtain the top $K_1$ index and sample $K_2$ from the rest, retrieve the potentials according to these indices;
4. correct the bias before the DP, then insert the corrected potentials to an existing DP implementation.

Now we explain each step in detail.

**Potential Normalization**   is a technique useful for numerical stability (see Fu et al., 2021). Specifically, in practice, the log-sum-exp function will saturate (it returns the max input value) when the maximum difference between inputs is larger than 10. So, we normalize all log potentials to be within range $[1, m], m < 10$. Given the emission potential $\phi$ and transition potential $\Phi$ from Eq. 8, we normalize as:

$$\tilde{\phi}(x_t, i) = m * \frac{\log \phi(x_t, i) - \min_i \log \phi(x_t, i)}{\max_i \log \phi(x_t, i) - \min_i \log \phi(x_t, i)} \tag{84}$$

$$\tilde{\Phi}(i, j) = m * \frac{\log \Phi(i, j) - \min_{i,j} \log \Phi(i, j)}{\max_{i,j} \log \Phi(i, j) - \min_i \log \Phi(i, j)} \tag{85}$$

Then the actual CRF is constructed based on the normalized potentials.

**Proposal Construction**   Our proposal is constructed based on the normalized local emission and a prior distribution. Specifically:

$$q_t(i) = \frac{1}{2} \cdot \left( \frac{\exp \tilde{\phi}(x_t, i)}{\sum_{i=1}^N \exp \tilde{\phi}(x_t, i)} + \frac{\exp ||\boldsymbol{s}_i||_1}{\sum_{i=1}^N \exp ||\boldsymbol{s}_i||_1} \right) \tag{86}$$

**Index Retrieval**   For each step $t$, we retrieve the top $K_1$ index and get a $K_2$ sized sample from the rest, as is discussed in the main paper:

$$[\sigma_{t,1}, ..., \sigma_{t,K_1}, ..., \sigma_{t,N}] = \arg \operatorname{sort}_i \{q_t(i)\}_{i=1}^N \tag{87}$$

$$[\delta_{t,1}, ..., \delta_{t,K_2}] \sim \text{Categorical}\{q_t(\sigma_{t,K_1} + 1), ..., q_t(\sigma_{t,N})\} \tag{88}$$

Note that for simplicity, we use sampling with replacement. For sampling *without* replacement, see the procedure in Kool et al. 2020

**Conventional Forward**   Before calling an existing implementation of Forward, we need to correct the bias of sampled terms. We do this by multiplying the probability of the sample to its emission potential (which becomes addition in log space). Note that this will be equivalent to multiplying them inside the DP:

$$\tilde{\phi}'(\sigma_{t,i}) = \tilde{\phi}(\sigma_{t,i}) \tag{89}$$

$$\tilde{\phi}'(\delta_{t,i}) = \tilde{\phi}(\delta t, i) + \log q_t(\delta_{t,i}) \tag{90}$$

We treat any duplicate indices as if they are different, and view the resulting potentials as if they are a new CRF. We also retrieve the transition potentials at each step according to the chosen index. So the new CRF has different transitions across steps. Finally, we run an existing efficient implementation of the Forward algorithm on the new CRF to get the estimated $\hat{\alpha}$ and $\hat{Z}$.

## D   EXTENSION OF RDP

In this section, we extend the RDP to more graph structures and more inference operations. We fisrt discuss a randomized inside algorithm for the partition function estimation for tree-structured hypergraphs, which includes dependency trees and PCFGs. Then we discuss a randomized entropy DP estimation on chain-structured graphs. This estimation will call the randomized forward as its subroutine. Finally we discuss howto generalize RDP to the general sum-product algorithm for any graph structure.

### D.1   EXTENSIONS TO TREE-STRUCTURED HYPERGRAPHS: RANDOMIZED INSIDE

This section discusses how to extend RDP to Tree-structured hypergraphs, including Dependency TreeCRFs and PCFGs. We focus on randomizing the Inside algorithm for partition function estimation. Specifically, the core recursion of the inside is:

$$\beta[i,j,k] = s_{ijk} \sum_{l=i}^{j-1} \sum_{k_1 \in \Omega, k_2 \in \Omega} \beta[i,l,k_1]\beta[l,j,k_2] \tag{91}$$

where $\beta[i,j,k]$ is the summation of all subtrees spanning from location $i$ to location $j$ with label $k$ and $s_{jik}$ is the local score and $\Omega$ is the full state space. Suppose $\Omega$ is large so we want to sample its subset $\Omega_{ij}$ to reduce the summation computation. Suppose the proposal is $q_{ijk}$ where $\sum_{k \in \Omega} q_{ijk} = 1$. Then a randomized inside recursion with $K_2$ sample is (we omit the top K1 summation for simplicity):

$$\beta[i,j,k] = s_{ijk} \sum_{l=i}^{j-1} \sum_{\delta_1 \in \Omega_{il}, \delta_2 \in \Omega_{lj}} \frac{1}{K_2 q_{il\delta_1}} \beta[i,l,\delta_1] \cdot \frac{1}{K2 q_{lj\delta_2}} \beta[l,j,\delta_2] \tag{92}$$

The analysis about bias and variance is similar to the previous analysis (Sec. A) on the linear-chain case.

### D.2   EXTENSIONS TO RANDOMIZED ENTROPY DP ESTIMATION

This section describes a randomized DP for estimating the entropy of a chain-structured model (HMMs and linear-chain CRFs). Recall that the core recursion of conventional entropy DP is:

$$p(z_t = i | z_{t+1} = j) = \frac{\Phi(i,j)\phi(x_{t+1},j)\alpha_t(i)}{\alpha_{t+1}(j)} \tag{93}$$

$$H_{t+1}(j) = \sum_{i=1}^{N} p(z_t = i | z_{t+1} = j)[H_t(i) - \log p(z_t = i | z_{t+1} = j)] \tag{94}$$

where $H_t(i)$ is the intermediate conditional entropy end at state $i$ step $t$. In our sampled DP, we first call the sampled forward to estimate the alpha variables. Then we re-use the sampled indices for the entropy DP graph, and the core recursion becomes:

$$\hat{p}(z_t = i | z_{t+1} = j) = \frac{\Phi(i,j)\phi(x_{t+1},j)\hat{\alpha}_t(i)}{\hat{\alpha}_{t+1}(j)} \tag{95}$$

$$H_{t+1}(j) = \sum_{\delta_t=1}^{K_2} \frac{1}{K_2 \cdot q_t(\delta_t)} \hat{p}(z_t = \delta_t | z_{t+1} = j)[H_t(i) - \log \hat{p}(z_t = \delta_t | z_{t+1} = j)] \tag{96}$$

where $q_t$ is the proposal at step $t$. Note how we re-use the estimated alpha variables $\hat{\alpha}$ for our entropy DP and correct the bias of each step.

### D.3   EXTENSIONS TO GENERAL SUM-PRODUCT

This section discusses the extension of sampled DP to general graph structures and message-passing algorithms, following the Bethe variational principle (Wainwright & Jordan, 2008). Recall the general message-passing algorithm computes pseudo marginals by recursively updating the message at

each edge:

$$M_{ts}(x_s) \leftarrow \kappa \sum_{x'_t} \left\{ \phi_{st}(x_s, x'_t)\phi_t(x'_t) \prod_{u \in N(t)/s} M_{ut}(x'_t) \right\} \tag{97}$$

where $M_{ts}(x_s)$ denotes the message from node $t$ to node $s$ evaluated at $X_s = x_s$, $\phi_{st}$ is the edge potential, $\phi_t$ is the node potential, $N(t)/s$ denotes the set of neighbor nodes of $t$ except $s$, and $\kappa$ the normalization constant. At convergence, we have the pseudo marginals

$$\mu_s(x_s) = \kappa\phi_s(x_s) \prod_{t \in N(s)} M^*_{ts}(x_s) \tag{98}$$

and the Bethe approximation of the log partition function is given by the evaluation of the Bethe variational problem:

$$\log Z_{\text{Bethe}} = \boldsymbol{\phi}^\mathsf{T}\boldsymbol{\mu} + \sum_s H_s(\mu_s) - \sum_{s,t} I_{st}(\mu_{st}) \tag{99}$$

where $H$ denotes marginal entropy and $I$ denotes marginal mutual information.

We consider the application of randomization to the computation of pseudo marginals and the Bethe log partition. Note that the Bethe approximation will be exact if the underlying graph is a tree. We consider a local proposal $\tilde{q}$ combined with a global prior $\tau$:

$$q(x_s) = \frac{1}{2}(\tilde{q}(x_s) + \tau(x_s)) \tag{100}$$

where the prior $\tau$ depends on our knowledge of specific problem structures. As one can always retreat to uniform proposals, we can explore more advanced choices. For $\tilde{q}$, one may consider: (a) local normalization as $\tilde{(q)}(x_s) = \phi(x_s)/\sum_s \phi(x_s)$ and (b) pre-computed mean-field approximations from algorithms like SVI (Hoffman et al., 2013). For $\tau$, one can further use a frequency-based empirical prior estimated from the data. To determine which nodes to perform DP on, one may consider the following principles:

- Special nodes of interest depending on their actual meaning. For example, one may be interested in some $x_s = 0$ (e.g., if a user is under a bank fraud threat).
- Nodes with large local weight $\phi(x_s)$.
- Nodes with large global weight $\tau(x_s)$.
- *Loop elimination*. If two nodes have similarly small local and global weight, we could drop those which eliminate loops in the resulting graph. Also note that we would prefer removing small-weighted nodes for loop elimination.

With the above principles, one may construct three subsets to perform the sum-product:

- $\Omega_1$ including nodes of special interest, where we perform exact computation.
- $\Omega_2$ from the top items of the proposal, where we also perform exact computation.
- $\Omega_3$ by sampling from the remaining items of the proposal (optionally with loop-reduction). For this set of nodes, one needs to correct the estimation by dividing the proposal probabilty: $\phi(x_s) \leftarrow \phi(x_s)/q(x_s)$.

After these steps, we treat nodes in $\Omega_1 \bigcup \Omega_2 \bigcup \Omega_3$ as if they are a new model, then feed them to an existing sum-product implementation.

## E  EXPERIMENT DETAILS

### E.1  MODEL ARCHITECTURE DETAILS

For training LVMs, partition function estimation, and paraphrasing, we use the Huggingface check-point of GPT2 model[1] since these experimental variables are more about autoregressive language

---

[1]https://huggingface.co/transformers/model_doc/gpt2.html

modeling and generation. For analyzing latent network topologies, we use Huggingface checkpoint of BERT base model[2] since it has been the main focus of most previous analytical work. The decoder LSTM is a one-layer LSTM with hidden state size 762, the same as the size of the contextualized embeddings. It shares the embeddings of the inferred states with the encoder, and uses its own input word embeddings, whose size is also 762. This architecture suffices for training LVMs and inferring latent networks. For paraphrase generation, we change the decoder to be conditional by: (a) using the average word embeddings of content words of the input sentence as its initial hidden state (rather than zero hidden states); (b) letting it attend to the embeddings of content words of the input sentence, and copy from them. This effectively makes this decoder conditional on the BOW of the content words of the input sentence. Then decoding a paraphrase becomes a neural version of slot filling: a sequence of latent states by traversing the network becomes a template of the sentence which we fill with words.

## E.2 DATASET PREPROCESSING

For the MSCOCO dataset, we use the GPT2 tokenizer to process sentences. As the official website [3] does not release a test split (only a training and a development split), we use the official development split as our test split, and re-split the official training split to be a new training and a development split. For training LVMs, we only use 1/10 of the training set for quicker experiments (the full dataset takes over more than 20 hours to converge). For paraphrasing, we use the full dataset.

For the 20News dataset, we use the data from the sklearn website[4]. We follow its official split and process the sentences with the BERT tokenizer.

## E.3 HYPERPARAMETERS AND TRAINING STRATEGIES

To get meaningful convergence of the latent space without posterior collapse, the following techniques are important: (a). set $\beta$ in the correct range. A large $\beta$ force the posterior to collapse to a uniform prior, while a small $\beta$ encourages the posterior to collapse to a Dirac distribution. (b). use word dropout in initial training epochs, otherwise the decoder may ignore the latent code. (c). use potential normalization, otherwise the logsumexp function used in the forward algorithm may saturate and only return the max of the input, this would consequently lead the posterior to collapse to few "activated" states and other states will never be inferred.

We further note that a full DP based entropy calculation will cause memory overflow with automatic differentiation. So we approximate it with local emission entropy, i.e., the sum of the entropy of the normalized local emission factors. Compared with the full entropy, this approximation does not directly influence the transition matrix, but encourages more activated states and mitigates the posterior collapse problem. To get a full entropy regularization, one can further regularize the transition matrix towards an all-one matrix, or extend our randomized DP to entropy computation. As the current local entropy regularization is already effective for inducing a meaningful posterior, we leave the full entropy computation to future work.

Our reported numbers are averaged over "good enough" runs. Specifically, for all hyperparameter configurations, we first run three random seeds and get the mean and standard deviation of the performance metrics. If the standard deviation is small enough, we report mean performance metrics (NLL, PPL, and iBLEU) and the standard deviation. If the standard deviation is large, we run extra five seeds. Then we drop runs with bad performance (usually 2-3), and compute the mean and standard deviation of the performance metrics again. In total, we experiment more than 100 runs over more than 7 hyperparameters: (a). learning rate $10^{-3}, 10^{-4}, 5 \times 10^{-4}$; (b). optimizer: SGD, Adam, AdamW; (c). dropout: 0.2, 0.3, 0.4; (d). $\beta$ variable: $10^{-4}, 5 \times 10^{-4}, 10^{-3}, 10^{-2}$; (e). Gumbel CRF v.s. Gumbel CRF straight-through (f). $K_1$ v.s. $K_2$; (g). word dropout schedule; and their combinations. Note that different models may achieve best performance with different hyperparameter combinations. We make sure that all models are searched over the same set of

---

[2]https://huggingface.co/transformers/model_doc/bert.html
[3]https://cocodataset.org/#home
[4]https://scikit-learn.org/stable/modules/generated/sklearn.datasets.fetch_20newsgroups.html

Table 3: Efficiency comparison with 100/ 500/ 2K states (Forward pass only).

| #States | 100 | | 500 | | 2000 | |
|---|---|---|---|---|---|---|
| Method | Mem | Time | Mem | Time | Mem | Time |
| FULL (Fu et al., 2020) | 3.9G | 0.24s | 7.9G | 1.07s | - | - |
| TOPK (Sun et al., 2019) | 4.1G* | 0.23s | 4.1G | 0.22s | 5.2G | 0.27s |
| RDP (ours) | 4.1G* | 0.24s | 4.1G | 0.22s | 5.2G | 0.27s |

hyperparameter combinations, and report average performance metrics over multiple seeds of the best hyperparameter configuration for each model.

### E.4 TIME COMPLEXITY ANALYSIS

Table 3 shows the actual efficiency comparison of our method. Time is measured by seconds per batch. For experiments with 500 and 2,000 states, we set $K = 100$. *When the number of states is small, our method underperforms due to the overhead of constructing the proposal. However, its advantage becomes clear as the number of states becomes large.

### E.5 VISUALIZATION PROCEDURE

This section describes how we produce Fig. 3(B1-2) and Fig. 4(A1-4). We use the sklearn implementation [5] of tsne (Van der Maaten & Hinton, 2008). For Fig. 3(B1-2), the word embeddings are obtained by sampling 8,000 contextualized embeddings from the full word occurrences in the 20News training set. Then we put the sampled word embeddings and the 2,000 states into the tsne function. The perplexity is set to be 30. An important operation we use, for better seperating the states from each other, it to manually set the distances between states to be large, otherwise the states would be concentrated in a sub-region, rather than spread over words. Fig. 4 is produced similarly, except we do not use word embeddings as background, and change the perplexity to be 5. For Fig. 4(A3), we connect the states if their transition potential is larger than a threshold. For Fig. 4(A4), we connect the states if their bigram frequency is larger than a threshold. All our decisions of hyperparameters are for the purpose of clear visualization which includes reducing overlapping, overcrowding, and other issues. We further note that no single visualization method can reveal the full structure of high-dimensional data, and any projection to 2-D plain inevitably induces information loss. We leave the investigation of better visualization methods to future work.

### E.6 STATE-WORD PAIR EXAMPLES

See Tables 8 and 9 for more sample.

### E.7 STATE-STATE TRANSITION EXAMPLES

See Table 10 for transitions involving stopwords and Table 11 for transitions without stopwords. Also note their differences as transitions involving stopwords are more about syntactic constructions and transitions without stopwords are more about specific meaning.

### E.8 STATE TRAVERSAL EXAMPLES

See Table 12 for more sample.

Table 4: Randomized Forward v.s. TopK Forward. Comparison of MSE.

| Method | Dense | Intermediate | Long-tail |
|---|---|---|---|
| TOPK 20% MEM (Sun et al., 2019) | 3.8749 | 1.0159 | 0.1629 |
| TOPK 50% MEM (Sun et al., 2019) | 0.99 | 0.2511 | 0.0315 |
| RDP 1% MEM (ours) | 0.1461 | 0.0669 | 0.0766 |
| RDP 10% MEM (ours) | 0.0672 | 0.0333 | 0.0552 |
| RDP 20% MEM (ours) | 0.0469 | 0.0200 | 0.0264 |
| RDP 50% MEM (ours) | 0.0078 | 0.0041 | 0.0046 |

Table 5: Randomized Forward, comparison of different proposal. Same base distribution as Figure 2, all proposals use 10% memeory).

| Method | Dense | Intermediate | Long-tail |
|---|---|---|---|
| UNIFORM (baseline) | 0.6558 | 11.0500 | 0.1600 |
| LOCAL ONLY (ours) | 2.908 | 9.1711 | 0.4817 |
| GLOBAL ONLY (ours) | 0.4531 | 0.0961 | 0.1003 |
| LOCAL + GLOBAL (ours) | 0.0284 | 0.0173 | 0.0222 |

# F    ADDITIONAL EXPERIMENT RESULTS

## F.1    RDP V.S. TOPK MSE COMPARISON

Table 4 shows the mean square error (MSE) comparison between RDP and TopK on dense, intermediate and long-tail distributions. Again, MSE for RDP is significantly smaller with less memory consumption.

## F.2    MSE COMPARISON OVER PROPOSALS

Table 5 shows the mean square error (MSE) comparison between different proposals. Our proposed local and global proposal outperforms a baseline uniform proposal on all distributions (dense, intermediate and long-tail).

## F.3    CONTROLLING NUMBER OF STATES

Figure 5 shows state frequency with different N. The long-tail behavior becomes clearer only when N is large enough (larger than 500 in our case).

## F.4    CONTROLLING $K_1$ V.S. $K_2$ RATIO

Figure 6 show state frequency with different $K_1/K_2$ ratios at different N. We highlight that when $K_2 = 0$, a pure topK summation approach would lead to posterior collapse where there exist inactive states that do not have any density. We also notice that an increasing $K_2$ consistently increases the frequency of tail states. This observation can be explained by the exploitation-exploration tradeoff, where increasing $K_1$ would encourage the exploitation of states already confident enough during training (consequently leading to high-frequency head states after convergence) while increasing $K_2$ would encourage exploring states that are not confident enough during training, leading to a larger tail frequency after convergence.

## F.5    COMPARISON TO RANDOMLY INITIALIZED BERT

Figure 7 shows the comparison of reconstruction NLL ($-\log p(x_t|z_t, \cdot)$) between a randomly initialized BERT and a pretrained BERT. States induced from pretrained BERT are more meaningful than random, making it easier to reconstruct words based on their corresponding states.

---

[5]https://scikit-learn.org/stable/modules/generated/sklearn.manifold.
TSNE.html

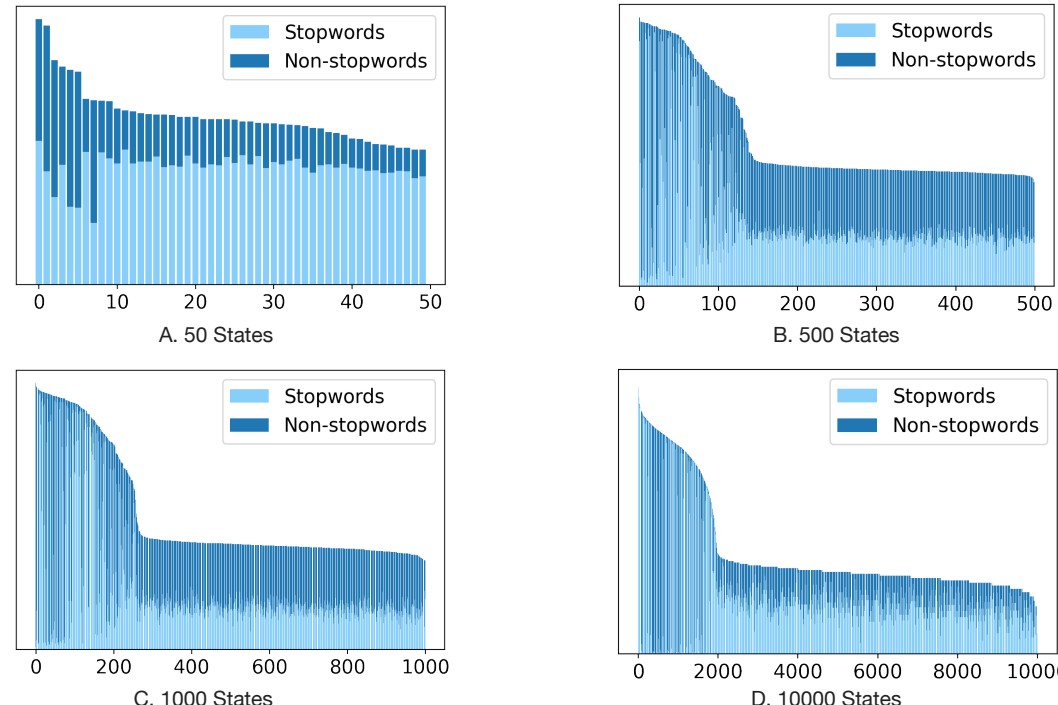

Figure 5: State frequency with different N (number of states). When N = 50, the long-tail behavior is not visible. The long-tail behavior emerges only when N is large enough (larger than 500 in our case).

Table 6: Randomized Inside algorithm for Tree-structured Hypergraphs. Comparison of MSE.

| Method | Dense | Intermediate | Long-tail |
|---|---|---|---|
| TOPK INSIDE 20% MEM | 36.1275 | 27.4351 | 21.7788 |
| TOPK INSIDE 50% MEM | 2.8422 | 2.4043 | 2.0479 |
| RANDOMIZED INSIDE 1% MEM (ours) | 26.3312 | 37.6698 | 48.8638 |
| RANDOMIZED INSIDE 10% MEM (ours) | 1.1937 | 1.5307 | 1.3843 |
| RANDOMIZED INSIDE 20% MEM (ours) | 0.4455 | 0.5449 | 0.5997 |

## F.6 RANDOMIZED INSIDE ALGORITHM FOR TREE-STRUCTURED HYPERGRAPH

We simulate three unit cases of treecrfs (dense, intermediate and long-tail) by controlling the entropy. We set the number of latent states to 2000. We use a uniform proposal for comparison, and set $K_1$ (top K sum size) = $K_2$ = 50. We use RDP with 20, 200, 400 states, which correspond to 1%, 10%, and 20% of the full number of states. We use topK summation as our baseline with 500 and 1000 states (25% and 50% of the full number of states). Figure 8 shows performance of RDP to tree structured hypergraph. Our method still outperforms topK summation (grey lines) with less memory (as our estimates are visually closer to the red line, i.e., the true partition). Table 6 shows the MSE of our method compared to topK summation, and the results are consistent with Figure 8.

## F.7 RANDOMIZED ENTROPY DP ALGORITHM

Figure 9 shows the application of RDP to entropy estimation of linear-chain CRFs. Our method consistently outperforms topK summation (grey lines) with less memory (as our estimates are visually closer to the red line, i.e., the true partition). Table 7 shows the MSE of our method compared to topK summation, and the results are consistent with Figure 9.

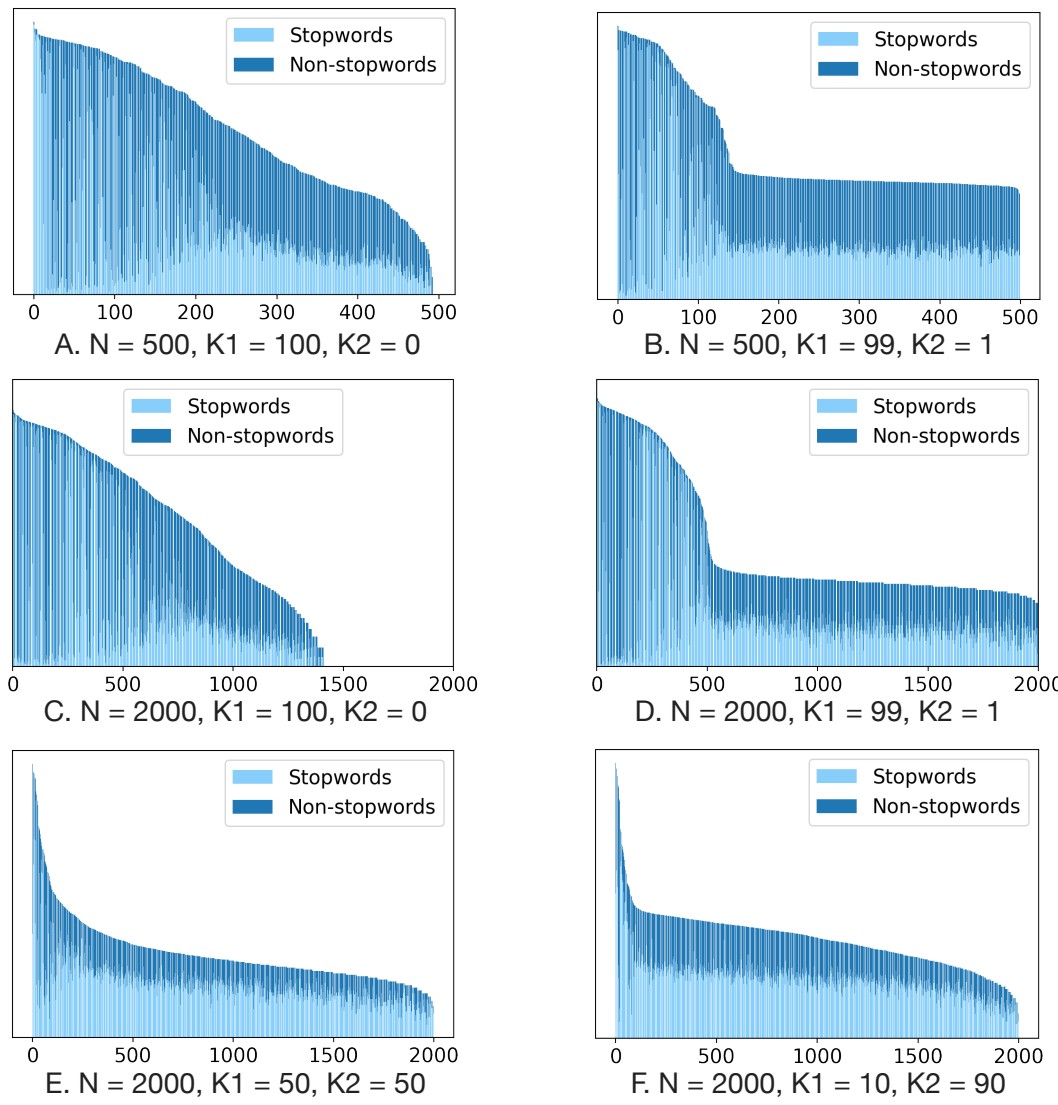

Figure 6: State frequency by controlling K1 (sum size) and K2 (sample size). We highlight that when $K_2 = 0$, a pure topK summation approach would lead to posterior collapse where there exist inactive states that do not have any density. We also notice that an increasing $K_2$ consistently increases the frequency of tail states.

Table 7: Randomized entropy DP v.s. TopK entropy DP on linear-chain CRFs. Comparison of MSE.

| Method | Dense | Intermediate | Long-tail |
|---|---|---|---|
| ENTROPY TOPK 20% MEM | 443.7 | 84.35 | 8.0115 |
| ENTROPY TOPK 50% MEM | 131.8 | 22.1 | 1.8162 |
| ENTROPY RDP 1% MEM (ours) | 5.9256 | 1.9895 | 0.6914 |
| ENTROPY RDP 10% MEM (ours) | 2.1168 | 1.2989 | 0.3167 |
| ENTROPY RDP 20% MEM (ours) | 1.3267 | 0.7305 | 0.2071 |
| ENTROPY RDP 50% MEM (ours) | 0.3017 | 0.1461 | 0.0631 |

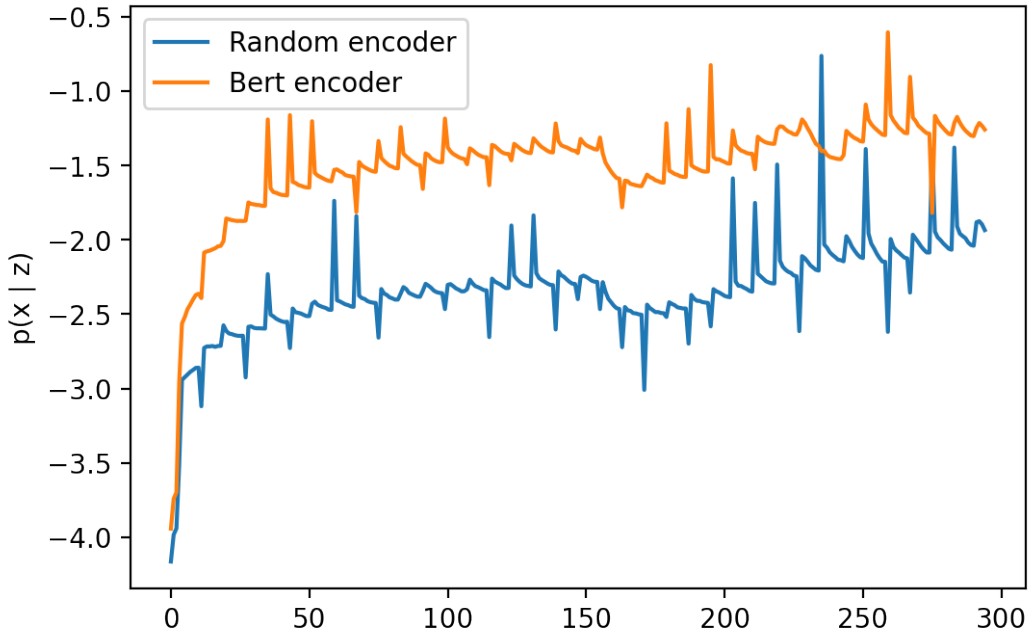

Figure 7: Comparison of reconstruction likelihood ($\log p(x_t|z_t, \cdot)$) between a randomly initialized BERT and a pretrained BERT. States induced from pretrained BERT are more meaningful than random, making it easier to reconstruct the words based on their corresponding states.

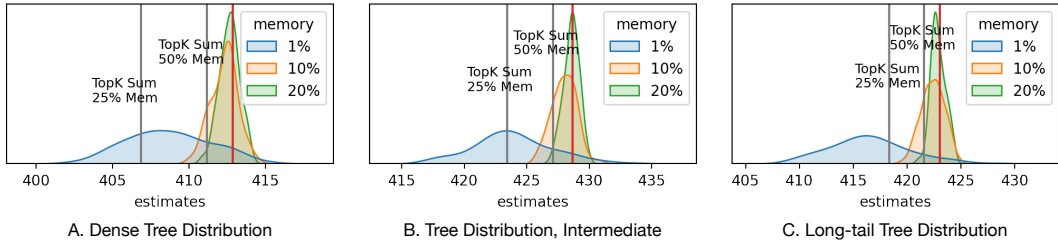

Figure 8: Application of RDP to tree structured hypergraph. Our method consistently outperforms topK summantion (grey lines) with less memory (as our estimates are visually closer to the red line, i.e., the true partition).

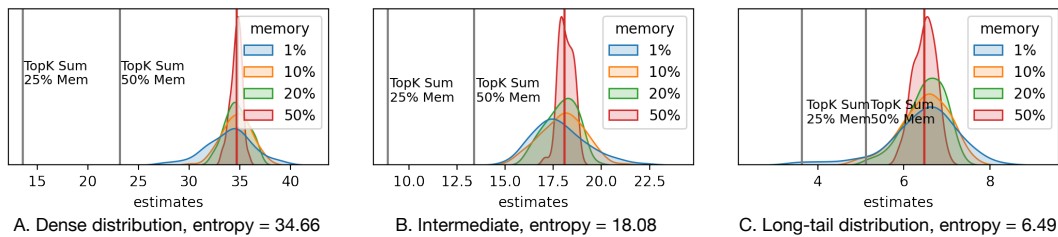

Figure 9: Application of RDP to entropy estimation of linear-chain CRFs. Our method consistently outperforms topK summation (grey lines) with less memory (as our estimates are visually closer to the red line, i.e., the true entropy).

| State id | Word - Occurrence |
|---|---|
| 1724 | ##s 5092 ; ##es 239 ; ##os 153 ; ##is 140 ; ##as 76 ; ##z 70 ; ##rs 54 ; ##gs 48 ; states 46 ; ##t 46 ; ##p 46 ; ##ps 43 ; ##ms 40 ; ##ns 35 ; ##e 34 ; ##r 31 ; ##i 29 ; ##ss 27 |
| 476 | ##ed 609 ; ##d 437 ; ##ted 156 ; based 144 ; caused 95 ; ##ized 75 ; made 61 ; lost 61 ; built 60 ; ##able 58 ; supported 57 ; expressed 56 ; occupied 54 ; defined 54 ; created 54 |
| 254 | david 521 ; john 404 ; mike 249 ; steve 245 ; dave 234 ; michael 223 ; robert 214 ; jim 204 ; mark 194 ; bob 173 ; paul 143 ; james 142 ; bill 133 ; tom 127 ; andrew 122 ; peter 121 |
| 710 | problem 599 ; killed 278 ; death 243 ; kill 235 ; problems 211 ; error 187 ; crime 181 ; murder 176 ; sin 175 ; genocide 172 ; evil 140 ; bad 129 ; hate 112 ; pain 110 ; massacre 97 |
| 1972 | ##s 649 ; turks 222 ; armenians 206 ; jews 186 ; keys 171 ; muslims 151 ; arabs 123 ; ##ms 122 ; christians 93 ; ##es 73 ; countries 71 ; villages 63 ; children 61 ; colors 61 ; guns 61 |
| 403 | god 1917 ; ##a 751 ; world 730 ; ca 347 ; country 292 ; usa 289 ; ##o 284 ; group 280 ; ##u 277 ; uk 244 ; groups 133 ; ##ga 126 ; ##g 125 ; europe 124 ; heaven 120 ; hell 119 |
| 1494 | problem 257 ; problems 87 ; agree 67 ; discussion 41 ; issue 39 ; argument 34 ; deal 34 ; issues 31 ; disagree 20 ; conflict 16 ; arguments 16 ; case 15 ; relationship 14 ; agreement 14 |
| 1419 | real 512 ; general 231 ; major 203 ; specific 157 ; main 156 ; actual 133 ; important 116 ; full 109 ; total 106 ; serious 99 ; absolute 88 ; basic 87 ; great 84 ; personal 83 ; true 76 |
| 305 | ##s 1336 ; opinions 340 ; problems 154 ; rules 140 ; views 113 ; things 104 ; laws 99 ; events 97 ; actions 94 ; issues 94 ; values 93 ; arguments 90 ; cases 82 ; rates 78 ; comments 78 |
| 555 | subject 2349 ; key 596 ; name 551 ; number 472 ; address 300 ; size 242 ; numbers 167 ; value 149 ; color 141 ; important 134 ; speed 122 ; names 110 ; level 106 ; count 84 |
| 1556 | ##ing 1282 ; running 188 ; ##ting 149 ; ##ng 104 ; ##ling 92 ; processing 87 ; killing 83 ; ##ding 71 ; calling 70 ; ##ring 69 ; getting 66 ; reading 61 ; ##izing 61 ; ##ping 60 |
| 472 | re 3084 ; com 2685 ; e 1034 ; ##e 892 ; internet 271 ; ##o 185 ; ##te 140 ; de 117 ; ##re 101 ; ##com 98 ; ##ne 71 ; ##r 71 ; ##er 70 ; org 70 ; ##me 70 ; net 68 ; se 62 |
| 1488 | human 344 ; turkish 255 ; political 246 ; moral 217 ; religious 186 ; legal 161 ; jewish 159 ; israeli 143 ; sexual 128 ; social 121 ; physical 119 ; personal 114 ; civil 104 ; international 103 |
| 1410 | simple 292 ; standard 213 ; normal 210 ; reasonable 208 ; fine 192 ; nice 191 ; interesting 138 ; good 128 ; correct 127 ; perfect 121 ; stupid 117 ; ##able 106 ; proper 105 ; true 91 |
| 243 | ##ly 929 ; probably 311 ; clearly 254 ; completely 231 ; obviously 229 ; certainly 222 ; directly 186 ; easily 179 ; apparently 176 ; generally 162 ; simply 154 ; necessarily 153 ; unfortunately 143 |
| 1418 | never 919 ; little 48 ; cannot 46 ; none 37 ; nobody 35 ; ever 31 ; nothing 27 ; without 25 ; neither 19 ; didn 14 ; ve 11 ; hardly 9 ; always 8 ; zero 8 ; m 7 ; non 6 ; doesn 6 |
| 1246 | ##t 1242 ; ##p 853 ; net 568 ; phone 395 ; call 381 ; bit 348 ; ##net 323 ; ##l 225 ; bat 135 ; line 123 ; ##g 111 ; ##d 101 ; network 94 ; ##it 93 ; tel 87 ; telephone 83 ; ##m 79 |
| 935 | long 682 ; hard 600 ; big 570 ; bad 518 ; good 483 ; great 404 ; fast 216 ; large 185 ; ##y 160 ; short 158 ; quick 154 ; little 150 ; strong 145 ; high 114 ; nice 113 ; hot 107 |
| 659 | cs 474 ; jp 468 ; cc 315 ; ms 291 ; cd 217 ; cm 194 ; dc 181 ; ##eg 158 ; bt 142 ; ps 125 ; ds 123 ; mc 120 ; pc 110 ; nc 110 ; gt 107 ; gs 107 ; rs 103 ; ss 103 ; bc 83 ; cb 81 |
| 1214 | space 182 ; nasa 170 ; orbit 169 ; motif 136 ; moon 103 ; planet 95 ; prism 94 ; lunar 92 ; venus 86 ; saturn 85 ; spacecraft 85 ; earth 80 ; shuttle 76 ; satellite 73 ; mars 70 |
| 654 | around 351 ; within 234 ; behind 137 ; across 102 ; regarding 81 ; among 73 ; throughout 70 ; beyond 57 ; near 53 ; along 53 ; inside 47 ; outside 44 ; past 40 ; concerning 38 ; following 37 |
| 127 | ##er 473 ; everyone 390 ; ##r 319 ; user 270 ; anyone 259 ; host 180 ; friend 161 ; one 150 ; server 136 ; everybody 109 ; fan 104 ; manager 103 ; nobody 101 ; player 93 ; guy 84 |
| 145 | believe 798 ; hope 421 ; evidence 379 ; claim 308 ; test 219 ; assume 201 ; proof 148 ; argument 145 ; prove 133 ; check 127 ; claims 106 ; suspect 104 ; doubt 95 ; guess 93 ; assuming 85 |

Table 8: More state-word examples

| State id | Word - Occurrence |
|---|---|
| 1572 | president 378 ; clinton 293 ; ##resh 268 ; myers 165 ; attorney 84 ; general 79 ; morris 78 ; smith 76 ; paul 75 ; bush 74 ; manager 64 ; hitler 56 ; ##ey 52 ; ##i 48 ; ##man 45 |
| 964 | cut 132 ; plug 121 ; break 73 ; thread 73 ; cable 63 ; hole 59 ; holes 54 ; chip 49 ; fix 48 ; clutch 48 ; stick 46 ; connector 42 ; blow 42 ; box 41 ; screw 40 ; pin 40 ; hit 40 |
| 1756 | see 1721 ; look 858 ; seen 618 ; read 302 ; saw 274 ; display 205 ; image 199 ; looks 197 ; looking 196 ; looked 188 ; screen 177 ; watch 161 ; view 153 ; monitor 149 ; images 132 |
| 585 | day 779 ; sun 686 ; ##n 556 ; today 310 ; night 276 ; week 269 ; days 264 ; city 161 ; morning 145 ; sunday 125 ; ##en 117 ; year 105 ; ##net 96 ; n 92 ; ##on 89 ; weeks 87 ; month 73 |
| 66 | power 433 ; control 399 ; state 347 ; virginia 202 ; mode 182 ; process 162 ; effect 139 ; period 118 ; action 117 ; authority 91 ; function 87 ; position 84 ; ##v 82 ; force 78 |
| 1240 | first 1443 ; always 602 ; ever 559 ; never 361 ; ago 321 ; often 319 ; sometimes 284 ; usually 203 ; early 195 ; last 192 ; every 175 ; later 171 ; soon 155 ; recently 146 ; past 145 |
| 865 | faith 383 ; religion 377 ; believe 211 ; atheist 205 ; ##ism 164 ; islam 159 ; ##sm 158 ; religious 145 ; morality 137 ; belief 126 ; font 115 ; language 114 ; truth 92 ; logic 90 |
| 467 | ##u 1203 ; point 784 ; happen 234 ; place 201 ; happened 183 ; colorado 141 ; happens 139 ; points 132 ; wait 126 ; ground 94 ; site 94 ; center 86 ; position 78 ; situation 78 ; 1993 76 |
| 203 | ca 273 ; pub 177 ; ##u 143 ; dod 143 ; au 141 ; mit 138 ; ma 132 ; ##si 129 ; sera 121 ; des 113 ; fi 75 ; isa 70 ; il 58 ; ny 58 ; po 56 ; la 53 ; tar 48 ; lee 47 ; ti 47 |
| 371 | drug 212 ; drugs 177 ; food 145 ; health 130 ; medical 121 ; disease 117 ; diet 115 ; cancer 113 ; aids 98 ; homosexuality 96 ; sex 83 ; homosexual 82 ; medicine 82 ; hiv 78 ; treatment 77 |
| 1683 | money 479 ; cost 307 ; pay 274 ; issue 212 ; problem 186 ; matter 175 ; worth 175 ; care 153 ; costs 146 ; tax 108 ; expensive 102 ; responsible 96 ; risk 96 ; spend 95 ; insurance 94 |
| 1856 | whole 379 ; entire 196 ; full 58 ; every 49 ; everything 42 ; together 37 ; everyone 25 ; ##u 17 ; rest 14 ; ##up 13 ; ##ed 13 ; away 10 ; always 10 ; top 10 ; open 10 ; ##s 9 |
| 1584 | university 1064 ; government 886 ; law 769 ; science 482 ; ##u 412 ; research 312 ; history 290 ; laws 165 ; study 125 ; policy 105 ; court 103 ; scientific 98 ; physics 97 ; constitution 93 |
| 1514 | life 663 ; live 363 ; security 192 ; exist 188 ; peace 180 ; dead 175 ; living 164 ; existence 157 ; body 157 ; lives 153 ; exists 137 ; privacy 128 ; death 126 ; die 121 ; safety 112 |
| 1208 | atheist 165 ; font 144 ; bio 119 ; bb 111 ; homosexual 109 ; ##group 99 ; sy 94 ; mormon 75 ; fed 72 ; manual 56 ; posting 52 ; spec 52 ; ##s 50 ; ##eri 49 ; auto 42 ; pointer 41 ; handgun 37 |
| 837 | another 974 ; last 774 ; next 581 ; else 578 ; second 498 ; others 472 ; third 124 ; first 82 ; final 69 ; later 60 ; rest 49 ; future 47 ; 2nd 44 ; latter 41 ; previous 40 ; elsewhere 33 |
| 1291 | lines 1848 ; read 701 ; writes 502 ; line 376 ; book 319 ; books 244 ; write 203 ; written 177 ; text 171 ; reading 157 ; wrote 144 ; article 107 ; quote 92 ; writing 86 ; paper 84 |
| 1656 | agree 182 ; solution 165 ; advice 128 ; opinion 110 ; interface 104 ; response 88 ; suggestions 80 ; recommend 75 ; alternative 75 ; discussion 71 ; offer 71 ; argument 70 ; application 69 |
| 1874 | apple 405 ; chip 386 ; disk 373 ; fbi 289 ; encryption 197 ; ##eg 171 ; hardware 166 ; nsa 154 ; ram 154 ; algorithm 134 ; tape 129 ; nasa 119 ; chips 111 ; ibm 100 ; floppy 98 |
| 1966 | stanford 269 ; washington 177 ; russian 156 ; cleveland 141 ; berkeley 137 ; california 131 ; chicago 105 ; ##co 96 ; turkey 95 ; york 83 ; boston 74 ; bosnia 73 ; soviet 71 ; russia 71 |
| 603 | file 682 ; list 526 ; article 501 ; card 424 ; bill 237 ; board 196 ; book 191 ; box 180 ; package 140 ; page 139 ; directory 119 ; section 118 ; group 114 ; library 90 ; files 83 |
| 1401 | done 644 ; didn 41 ; perform 35 ; performed 32 ; accomplish 25 ; accomplished 16 ; could 14 ; ##d 11 ; conduct 10 ; happen 10 ; say 10 ; committed 9 ; finish 9 ; completed 9 ; conducted 8 |
| 460 | clip 186 ; ##op 175 ; com 162 ; news 162 ; posts 109 ; works 106 ; micro 68 ; sim 66 ; share 66 ; ##yp 58 ; net 58 ; wire 54 ; ##os 48 ; power 43 ; es 40 ; flop 39 ; mac 39 ; tool 39 |

Table 9: More state-word examples, continued.

| Transition | Bigram - Occurrence |
| --- | --- |
| 1843-990 | is-that 68 ; fact-that 51 ; so-that 50 ; think-that 46 ; note-that 41 ; say-that 39 ; sure-that 38 ; believe-that 35 ; out-that 33 ; know-that 32 ; seems-that 28 ; mean-that 26 |
| 1010-1016 | instead-of 40 ; amount-of 33 ; lot-of 26 ; form-of 23 ; lack-of 19 ; institute-of 16 ; case-of 15 ; capable-of 14 ; amounts-of 13 ; out-of 12 ; years-of 12 ; department-of 11 ; terms-of 11 |
| 960-458 | up-to 56 ; down-to 29 ; access-to 25 ; according-to 24 ; due-to 22 ; go-to 17 ; response-to 14 ; subject-to 13 ; related-to 13 ; reference-to 13 ; as-to 12 ; lead-to 12 ; reply-to 12 |
| 441-698 | have-to 139 ; going-to 116 ; seem-to 114 ; seems-to 68 ; supposed-to 40 ; need-to 30 ; had-to 26 ; used-to 22 ; want-to 20 ; seemed-to 15 ; tend-to 14 ; appears-to 13 ; likely-to 13 ; appear-to 12 |
| 1712-698 | trying-to 67 ; try-to 46 ; able-to 43 ; like-to 41 ; hard-to 32 ; seem-to 22 ; seems-to 22 ; want-to 21 ; tend-to 21 ; willing-to 18 ; tried-to 16 ; enough-to 14 ; attempt-to 13 ; continue-to 12 |
| 1814-1666 | about-it 91 ; of-it 71 ; with-it 42 ; to-it 29 ; for-it 27 ; do-it 26 ; on-it 24 ; have-it 12 ; understand-it 11 ; doing-it 9 ; know-it 9 ; see-it 8 ; call-it 8 ; believe-it 8 ; ##ing-it 7 ; fix-it 6 |
| 1295-523 | problem-with 32 ; deal-with 32 ; do-with 28 ; up-with 16 ; problems-with 15 ; came-with 13 ; comes-with 13 ; along-with 12 ; work-with 12 ; contact-with 11 ; wrong-with 10 ; agree-with 10 ; disagree-with 9 |
| 628-150 | based-on 65 ; depending-on 23 ; is-on 13 ; ##s-on 11 ; down-on 9 ; effect-on 9 ; are-on 8 ; working-on 8 ; effects-on 7 ; activities-on 7 ; depend-on 7 ; be-on 6 ; run-on 6 ; depends-on 6 |
| 477-1414 | have-to 117 ; going-to 45 ; is-to 37 ; had-to 32 ; decided-to 12 ; need-to 11 ; has-to 11 ; having-to 9 ; required-to 9 ; willing-to 8 ; how-to 8 ; ,-to 7 ; reason-to 7 ; forced-to 7 |
| 477-1277 | is-to 74 ; have-to 43 ; had-to 20 ; used-to 17 ; required-to 14 ; going-to 14 ; ,-to 13 ; need-to 13 ; as-to 12 ; order-to 11 ; needed-to 11 ; ##s-to 10 ; be-to 10 ; decided-to 10 |
| 145-461 | believe-that 70 ; claim-that 24 ; evidence-that 18 ; assume-that 17 ; hope-that 15 ; belief-that 11 ; sure-that 9 ; prove-that 9 ; assuming-that 8 ; argue-that 8 ; likely-that 7 ; claims-that 7 |
| 278-217 | know-of 22 ; end-of 16 ; out-of 14 ; think-of 13 ; ##s-of 10 ; accuracy-of 8 ; top-of 7 ; friend-of 6 ; copy-of 6 ; heard-of 6 ; one-of 4 ; middle-of 4 ; version-of 4 ; beginning-of 4 ; aware-of 4 |
| 1820-276 | come-out 30 ; came-out 17 ; coming-out 14 ; put-out 12 ; get-out 11 ; find-out 10 ; check-out 9 ; turns-out 7 ; found-out 7 ; turn-out 7 ; turned-out 7 ; comes-out 7 ; go-out 6 ; ##ed-out 6 |
| 1142-461 | is-that 17 ; fact-that 15 ; understand-that 12 ; see-that 11 ; realize-that 11 ; noted-that 8 ; says-that 8 ; note-that 7 ; read-that 7 ; forget-that 6 ; out-that 6 ; shows-that 6 |
| 1010-1998 | lot-of 34 ; set-of 26 ; bunch-of 24 ; lots-of 22 ; series-of 13 ; number-of 10 ; thousands-of 10 ; hundreds-of 10 ; plenty-of 10 ; full-of 7 ; pack-of 7 ; list-of 6 ; think-of 5 |
| 1125-843 | of-a 124 ; is-a 86 ; for-a 84 ; to-a 50 ; s-a 16 ; be-a 14 ; ,-a 11 ; as-a 7 ; was-a 5 ; on-a 5 ; with-a 4 ; am-a 3 ; about-a 3 ; in-a 2 ; into-a 2 ; were-a 2 ; its-a 1 ; surrounding-a 1 |
| 476-1654 | written-by 13 ; ##d-by 11 ; caused-by 8 ; ##ed-by 8 ; produced-by 6 ; followed-by 6 ; defined-by 4 ; committed-by 4 ; hit-by 4 ; supported-by 4 ; led-by 4 ; explained-by 4 ; run-by 4 |
| 1812-837 | the-other 86 ; the-next 77 ; the-last 62 ; the-second 48 ; the-first 14 ; the-latter 10 ; the-third 9 ; the-latest 7 ; the-rest 6 ; the-previous 6 ; the-final 5 ; the-fourth 3 ; the-nearest 3 |
| 1938-145 | i-believe 128 ; i-hope 66 ; i-suspect 28 ; i-assume 24 ; i-doubt 18 ; i-suppose 11 ; i-guess 11 ; i-expect 8 ; i-think 7 ; i-imagine 6 ; i-feel 5 ; i-trust 4 ; i-gather 3 ; i-bet 2 |
| 1820-1856 | pick-up 14 ; come-up 12 ; came-up 11 ; stand-up 11 ; set-up 11 ; bring-up 8 ; show-up 8 ; comes-up 7 ; screwed-up 7 ; give-up 6 ; wake-up 6 ; speak-up 5 ; look-up 5 ; back-up 5 |
| 1417-979 | more-than 163 ; better-than 33 ; less-than 13 ; faster-than 12 ; greater-than 11 ; longer-than 8 ; ##er-than 7 ; larger-than 6 ; worse-than 6 ; higher-than 6 ; slower-than 6 ; easier-than 4 |
| 111-111 | of-the 75 ; to-the 34 ; for-the 23 ; on-the 14 ; with-the 12 ; about-the 7 ; part-of 7 ; in-the 5 ; into-the 5 ; like-the 4 ; out-of 4 ; at-the 4 ; by-the 3 ; '-s 3 ; as-the 2 |
| 1579-654 | talking-about 45 ; talk-about 25 ; concerned-about 14 ; worried-about 9 ; know-about 8 ; stories-about 7 ; worry-about 7 ; talked-about 6 ; rumours-about 5 ; news-about 5 ; feel-about 5 ; care-about 4 |

Table 10: State transition examples, with stopwords

| Transition | Bigram - Occurrence |
|---|---|
| 371-371 | health-care 14 ; side-effects 8 ; im-##mun 4 ; infectious-diseases 4 ; yeast-infections 4 ; ##thic-medicine 3 ; treat-cancer 3 ; health-insurance 3 ; barbecue-##d 3 ; hiv-infection 3 ; yeast-syndrome 3 |
| 1214-1214 | orbit-##er 14 ; astro-##physics 7 ; lunar-orbit 7 ; space-shuttle 7 ; earth-orbit 5 ; pioneer-venus 5 ; space-station 5 ; space-##lab 4 ; lunar-colony 4 ; orbit-around 3 ; space-tug 3 ; space-##flight 3 |
| 716-1556 | mail-##ing 15 ; fra-##ering 12 ; ##mina-##tion 9 ; bash-##ing 7 ; ##dal-##izing 6 ; ##ras-##ing 5 ; ##band-##ing 4 ; ##ress-##ing 4 ; cab-##ling 4 ; adapt-##er 4 ; cluster-##ing 4 ; sha-##ding 4 |
| 931-931 | gamma-ray 17 ; lead-acid 9 ; wild-corn 4 ; mile-long 3 ; smoke-##less 3 ; drip-##py 2 ; diamond-stealth 2 ; cold-fusion 2 ; 3d-wire 2 ; acid-batteries 2 ; schneider-stealth 2 ; quantum-black 2 |
| 1488-1488 | law-enforcement 17 ; national-security 5 ; cold-blooded 4 ; health-care 4 ; human-rights 4 ; im-##moral 4 ; prophet-##ic 4 ; social-science 3 ; ethnic-##al 3 ; turkish-historical 3 |
| 1246-1246 | bit-##net 35 ; tel-##net 12 ; use-##net 7 ; phone-number 7 ; dial-##og 6 ; ##p-site 5 ; phone-calls 5 ; bit-block 5 ; net-##com 4 ; bat-##f 4 ; ##t-##net 4 ; phone-call 4 ; arc-##net 3 |
| 1556-1556 | abu-##sing 5 ; ##dal-##izing 4 ; obey-##ing 4 ; robb-##ing 3 ; ##ov-##ing 3 ; dial-##ing 3 ; contend-##ing 3 ; ##upt-##ing 3 ; rough-##ing 3 ; contact-##ing 3 ; bash-##ing 3 ; favor-##ing 2 |
| 202-202 | western-reserve 21 ; case-western 20 ; ohio-state 19 ; united-states 10 ; penn-state 5 ; african-american 5 ; north-american 5 ; middle-eastern 5 ; polytechnic-state 4 ; north-carolina 4 |
| 1912-1912 | world-series 9 ; home-plate 7 ; division-winner 4 ; runs-scored 4 ; batting-average 4 ; game-winner 3 ; sports-##channel 3 ; plate-umpire 3 ; baseball-players 3 ; league-baseball 3 |
| 1461-1461 | ##l-bus 5 ; bit-color 5 ; 3d-graphics 4 ; ##p-posting 3 ; computer-graphics 3 ; wire-##frame 3 ; bit-graphics 2 ; ##eg-file 2 ; access-encryption 2 ; ##frame-graphics 2 ; file-format 2 |
| 123-123 | health-care 10 ; high-school 6 ; es-##crow 6 ; key-es 5 ; high-power 4 ; local-bus 4 ; low-level 4 ; high-speed 3 ; minor-league 2 ; health-service 2 ; regular-season 2 ; mother-##board 2 |
| 1702-1702 | mile-##age 8 ; engine-compartment 5 ; semi-auto 5 ; manual-transmission 5 ; drive-power 4 ; door-car 3 ; passenger-cars 3 ; sports-car 3 ; shaft-drive 3 ; mini-##van 3 ; speed-manual 3 |
| 1874-1874 | floppy-disk 11 ; jp-##eg 11 ; encryption-algorithm 8 ; ##per-chip 7 ; ##mb-ram 7 ; ##ga-card 6 ; encryption-devices 5 ; silicon-graphics 4 ; disk-drive 4 ; floppy-drive 4 |
| 1208-1064 | atheist-##s 43 ; homosexual-##s 12 ; fed-##s 9 ; libertarian-##s 8 ; ##eri-##s 7 ; ##tile-##s 7 ; azerbaijani-##s 6 ; ##tar-##s 6 ; mormon-##s 5 ; sniper-##s 5 ; physicist-##s 4 |
| 1710-1710 | power-supply 5 ; atomic-energy 4 ; water-ice 4 ; power-cord 4 ; ##com-telecom 3 ; light-pollution 3 ; light-bulb 3 ; radio-station 3 ; radio-##us 3 ; air-conditioning 3 ; light-##wave 2 |
| 1080-1080 | public-access 19 ; via-anonymous 5 ; private-sector 5 ; available-via 4 ; general-public 4 ; community-outreach 4 ; public-domain 3 ; personal-freedom 3 ; private-property 3 ; private-activities 3 |
| 254-1572 | jimmy-carter 9 ; george-bush 9 ; bill-clinton 8 ; bryan-murray 4 ; joe-carter 4 ; henry-spencer 4 ; bill-james 4 ; janet-reno 4 ; craig-holland 4 ; clayton-cramer 4 ; ##zie-smith 4 |
| 1571-1571 | ms-windows 24 ; windows-nt 12 ; ibm-pc 10 ; ms-##dos 7 ; unix-machine 6 ; microsoft-windows 5 ; windows-applications 4 ; run-windows 3 ; apple-monitor 3 ; mac-##s 3 ; desktop-machine 3 |
| 66-66 | ##ian-1919 3 ; energy-signature 2 ; charlotte-##sville 2 ; environment-variables 2 ; duty-cycle 2 ; second-period 2 ; spin-state 2 ; power-consumption 2 ; inter-##mission 2 ; power-play 2 |
| 1683-1683 | worth-##while 4 ; nominal-fee 4 ; get-paid 3 ; risk-factors 3 ; scholarship-fund 2 ; cost-\$ 2 ; tax-dollars 2 ; beneficial-item 2 ; bank-account 2 ; take-responsibility 2 |
| 1579-1579 | m-sorry 5 ; news-reports 4 ; heard-anything 4 ; ran-##ting 3 ; short-story 3 ; news-reporters 3 ; press-conference 3 ; heard-something 3 ; tv-coverage 2 ; horror-stories 2 ; heard-horror 2 |
| 1656-1656 | urbana-champaign 3 ; peace-talks 3 ; acceptable-solutions 2 ; marriage-partner 2 ; intercontinental-meetings 2 ; interested-parties 2 ; conference-calls 2 ; handle-conference 2 ; cooperative-behaviour 2 |

Table 11: State transition examples, without stopwords

| State Sequences | Sentence |
| --- | --- |
| 1755 0 117 1755 1755 0 117 1138 1755 103 117 1138 1755 117 | a white bathroom with a yellow toilet and a bath tub and a sink |
| 1755 0 117 1755 1755 959 117 117 1138 0 103 117 | a white bathroom with a yellow toilet sink and painted bath tub |
| 1755 117 1755 1755 0 117 1138 0 117 1138 103 117 | a bathroom with a yellow toilet and white sink and bath tub |
| 1755 117 0 1755 117 441 959 0 1061 1755 117 | a man reading a newspaper on the side of a road |
| 1755 117 0 1755 1755 441 959 117 1061 1755 117 | a man reading a newspaper on the side of a road |
| 1755 117 441 959 117 1061 1755 117 0 959 103 959 | |
| a man on the side of a road reading the newspaper | |
| 1755 117 0 1061 1755 117 1138 0 0 117 | a man posing for a camera while holding filthy bananas |
| 1755 117 0 1061 959 103 0 117 | a man posing for the camera holding bananas |
| 1755 117 0 0 1755 117 1138 0 959 103 103 959 | a man posing for a camera while holding his filthy bananas |
| 1755 0 103 117 1061 1755 0 117 1061 1755 103 | a silver clock tower under a black intersection near a tree |
| 1755 0 103 117 441 1755 0 117 1061 1755 103 | a silver clock tower on a black intersection near a tree |
| 103 117 1755 959 103 1061 0 103 103 103 | clock tower in the intersection of black near silver tree |
| 1755 103 103 117 1755 1755 117 1755 1755 0 117 1061 1755 117 | a motor cycl ist in a photograph with a blur motion of a freeway |
| 1755 117 1061 1755 0 103 103 117 959 1755 1755 1755 1755 1061 117 117 | a photograph of a motor cycl ist motion , with a blur in the freeway |
| 103 103 103 1755 0 0 959 117 1061 103 103 117 | motor cycl ist in motion blur the freeway at motor cycl ist |
| 1755 117 1061 103 103 103 1755 117 0 1061 1755 117 1755 0 117 | a piece of chocolate bread on a cart spread to a plate with dark bananas |
| 1755 117 1061 0 103 103 1755 117 1138 1017 103 103 0 527 1755 1755 117 | a piece of dark chocolate bread with bananas and a on cart spread in a plate |
| 103 103 1755 117 117 441 0 117 441 117 | chocolate cart with bananas spread on dark cart on plate |
| 1755 117 0 1755 117 1061 117 1755 1755 0 117 | a lot filled with lots of flowers in a v ases |
| 1755 117 0 1755 117 1061 117 959 1138 903 117 1061 1687 | a lot filled with lots of flowers , and v ases of it |
| 1755 117 1061 1082 117 0 1755 1138 1061 1595 959 | a lot of v ases filled with lots of flowers |
| 0 117 0 1755 117 1755 959 103 441 1755 103 1755 117 1755 959 117 | two people walking with dogs in the ocean on a beach with people in the background |
| 0 117 441 103 1755 117 1755 117 959 | two people on beach with dogs in background |
| 0 117 117 1755 117 441 959 1755 1755 959 117 | two people walking with dogs on the beach in the background |
| 1755 117 1061 117 0 1952 0 1061 1755 0 103 | a couple of women kneeling down next to a blow cat |
| 1755 103 0 1952 1061 1061 1755 0 0 117 1755 1755 0 103 | a cat kneeling down next to a couple blow women in a dry er |
| 1755 117 1061 117 0 1952 0 1061 1755 103 0 103 | a couple of women kneeling down next to a blow dry er |
| 1755 117 0 1755 0 103 441 1755 117 | a hand holds a hot dog on a table |
| 1755 117 117 1755 0 103 103 959 103 | a hand holds a hot dog in the table |
| 1755 117 0 1755 0 103 1061 1755 117 959 | a hand holds a hot dog at a table |
| 1755 117 0 1755 103 1755 1755 117 1061 117 117 | a woman eating a pastry in a piece of market area |
| 1755 117 1017 150 1113 0 1061 103 1061 1755 117 117 | a woman is eating a piece of pastry at a market area |
| 117 0 1755 103 1061 1755 0 1061 117 117 | woman eating a pastry at a piece of market area |

Table 12: Paraphrase Examples.

