# OpenReview forum: "Discovering Latent Network Topology in Contextualized Representations with Randomized Dynamic Programming"
_ICLR.cc/2022/Conference — ICLR 2022 Submitted_

### Official Review · Reviewer_Cp5o · 2021-10-29

**Correctness:** 4
**Technical Novelty And Significance:** 3
**Empirical Novelty And Significance:** 3
**Recommendation:** 6
**Confidence:** 4

**Main Review:**

# Summarization

In this paper, the authors address the issue in the computational and memory efficiency in Dynamic Programming when it is used in the Forward Algorithm for Linear Chain CRF. The author purpose  a Sample Forward Algorithm by randomizing the summation. The authors proves that such randomization is an unbiased estimator for the true sums, and, using Amortized Variational inference for optimization, shows that it empirically reveals meaningful latent topology with much less required resource.

# Strong Points

Generally this paper is an interesting one, well motivated and easy to follow. It combines the randomized summation, an idea used in [1], to the problem of Forward Algorithm, by viewing Forward Algorithm as a sum-product algorithm applied over the consecutive time steps. Such randomization is proved by the authors to be an unbiased estimator for the true sums.  The experimental section focuses on showing the meaningfulness of the learned latent space, mostly through state-world/state-state relations.

Note that the author uses recently amortized variational inference [1] for optimization and LSTM for parameterization of the forward algorithm [2], which are good to have, yet not central to the main contribution.

Although the authors frame it in Linear Chain CRF problems with pre-trained language model for contextualization, I think it could be conceptually expected to a broader spectrum of sum-product algorithms (the author briefly covered that in the Appendix), although this may be beyond the scope of a single paper.

# Weak Points:

There are a few missing pieces in this work which should be better addressed.

The first is the difference between the true proposal $\~{a}$ that is proved by the authors to lead to an unbiased estimator but cannot achieve improvements in computational efficiency, and the practically used proposal as in Eq. 10.  Although Fig 2. shows some distributions by using the practically used prospal that show no bias, it remains unclear how much it differs from the true proposal. Maybe a study of distances vs. memory presentation (e.g., $K_1$) could show a better picture of the trade-offs here.

The second issue is that, the conceptual benefit from the proposed method is being able to deal with larger $N$ (the number of latent states) where full Dynamic Programming could not run efficiently. However there is no such experiment showing what would be happen with larger $N$.

Finally, the experiments visualizing the latent space are for the proposed method only. It would be interesting to see how it compares with methods that only takes top $K_1$ and the vanilla Forward Algorithm (which takes the full $N^2$ transition matrix)

# Questions for the Authors

  - How are interpretations of states in experiments (e.g. in Fig. 3 and 4) coming from? Are they chosen in some particular way?
  - Maybe “the more $q_i$ correlates with $a_i$, the less variance $\hat{S}_2$ has“ can be explained in more detail ?
  - In Eq 11: $[s_{z_{t-1}; x_{t-1}}]$ should be $[s_{z_{t-1}; r_{t-1}}]$?
  - In Figure 3, subfigure B1 and B2, states and words are shown in the same space. What are exactly the features for words and states ($r$? $s$? or something else?)?


# Assessment
I think this is an interesting paper and is above the threshold of acceptance since the presented method is somewhat novel and inspiring. I’m willing to increase my score if my concerns are addressed

# Refs

[1] Kool et. al. 2020: Estimating Gradients for Discrete Random Variables by Sampling without Replacement
[2] Fu et al. 2020:  Latent Template Induction with Gumbel-CRFs
[3] Li & Rush, 2020: Posterior Control of Blackbox Generation


# Post-rebuttal

I thanks the authors for the responses which improves the manuscript a lot. Still, I would like to keep my recommendation "6: marginally above the acceptance threshold", since (1) the contribution on RDP and that in linguistic application are entangled and do not provide a clear message, and (2) as issues in my reviews and other reviews remain.


**Summary Of The Paper:**

In this paper, the authors address the issue in the computational and memory efficiency in Dynamic Programming when it is used in the Forward Algorithm for Linear Chain CRF. The author purpose  a Sample Forward Algorithm by randomizing the summation. The authors proves that such randomization is an unbiased estimator for the true sums, and, using Amortized Variational inference for optimization, shows that it empirically reveals meaningful latent topology with much less required resource.


**Summary Of The Review:**

I think this is an interesting paper and is above the threshold of acceptance since the presented method is somewhat novel and inspiring. I’m willing to increase my score if my concerns are addressed

---

> ### Author Response · Authors · 2021-11-15
> **Response to Reviewer 4 (Cp5o) (2/2)**
>
> ## Comparison to topK summation induced latent space
>
> - Firstly, we consider the more general question: what would happen if we control the ratio of K1 (sum size) and K2 (sample size). When we set K2 = 0, then this would amount to a topK only approach. We also note that we can only set K1 + K2 = 100 at maximum, which means that exact Forward computation for 500 states is still infeasible.
> - We have added new experimental results on controlling K1 v.s. K2 ratio in the appendix F.4. Specifically, we compare:
>     - Full number of states = 500:
>         - K1 =100, K2 = 0. This amounts to topK summation only. In this case we observe posterior collapse: certain states (about 20) become inactive and will not be inferred.
>         - K1=99, K2=1. This only adds one single sample, but immediately overcomes the posterior collapse problem. All states are active, and a long-tail distribution is formed. The reason that sampling helps posterior collapse can be explained by latent space exploration, since sampling means to explore (=pass gradients to) the states that are not confident enough.
>     - Full number of state = 2000:
>         - K1 = 100, K2 = 0: We experience more severe posterior collapse as about 700 states become inactive. This is because the topK summation approach only exploits high-confident states during training.
>         - Then we increase the ratio of K2 to be: (a) K1 = 99, K2 = 1; (b) K1 = 50, K2 = 50; (c) K1 = 10, K2 = 90; As we increase the ratio of K2, the frequency of tail states consistently increases, i.e., "tails" become more frequently observed. This is also consistent with our exploration v.s. exploitation argument: more sampling would encourage more exploration of the less prominent states, thus preventing posterior collapse to most frequent states.
>
> ## Detailed variance analysis, an asymptotic argument
>
> - We have expanded the argument “the more qi correlates with ai, the less variance S^2 has“ to a full variance analysis in appendix A.2. Specifically:
>     - In Theorem A.2, for the simple estimator in equation 2, we prove its variance is proportional to
>         - The squared sum of the tail, which can be reduced by increasing K1, as the effect of Rao-Blackwellization (the square comes from the nature of variance)
>         - The L2 distance from qi and ai, which can be decreased by decreasing the distance, as the effect of importance sampling.
>         - The sample size K2, as the effect of the central limit theorem
>     - In Theorem A.3, for the alpha recursion in equation 6, we prove the variance is asymptotically proportional to
>         - A tail sum term, which can be reduced by increasing K1, as the effect of Rao-Blackwellization.
>         - An error term between our proposal and the oracle proposal, which can be reduced by a more correlated proposal, as the effect of importance sampling.
>         - The sample size K1, as the effect of the central limit theorem.
>     - In Theorem A.4, for the partition estimator in equation 7, we prove variance is asymptotically proportional to:
>         - A cumulated tail sum term at each step, which can be reduced by increasing K1, as the effect of Rao-Blackwellization.
>         - A cumulated error term at each step between our proposal and the oracle proposal, which can be reduced by a more correlated proposal, as the effect of importance sampling.
>         - The sample size K1, as the effect of the central limit theorem.
>     - The above three arguments together give a full variance analysis of all the estimators proposed by this paper. Please refer to Appendix A.2 for details on specific equations and derivations.
>
> ## More questions
>
> In Eq 11: [szt−1;xt−1] should be [szt−1;rt−1]?
>
> - Not really. The decoder does not use the contextualized representation of the word (these embeddings are not static). It has its own word embedding matrix (this is why we can use it for paraphrase generation). Yet it does share the embeddings of the inferred states with the encoder to receive the state information.
>
> In Figure 3, subfigure B1 and B2, states and words are shown in the same space. What are exactly the features for words and states (r? s? or something else?)?
>
> - For words, we use r, i.e., their contextualized embeddings (the output of BERT's last layer)
> - For states, we use s.
> - We assume s is the unobserved latent states within the space of r, we infer s, and jointly visualize s and r.

---

> ### Author Response · Authors · 2021-11-15
> **Response to Reviewer 4 (Cp5o) (1/2)**
>
> We thank the reviewer for the detailed review. Here is our response:
>
> ## Distance v.s. Memory Study
>
> - Thank you for pointing this out. Distances from the true value can be visually inspected by looking at the distance from the red line in Figure 2. Specifically, the standard deviation of our method (width of the curve) is smaller than the distance from the grey line (topK estimate) to the red line (true value).
> - As an additional result, we provide the mean square error (recall that MSE is equal to L2 distance, and can be decomposed as bias^2 + variance) of different estimators here:
>
>
> | Method | Dense | Intermediate | Long-tail |
> | --------- | ------- | -------------- | -----------|
> | TopK 20% Mem | 3.8749 | 1.0159 | 0.1629 |
> | TopK 50% Mem | 0.99 | 0.2511 | 0.0315 |
> | RDP 1% Mem | 0.1461 | 0.0669 | 0.0766 |
> | RDP 10% Mem | 0.0672 | 0.0333 | 0.0552 |
> | RDP 20% Mem |  0.0469 | 0.0200 | 0.0264 |
> | RDP 50% Mem | 0.0078 | 0.0041 | 0.0046 |
>
> - We can see that our RDP generally outperforms TopK summation for all underlying distributions with significantly less memory (e.g, RDP with 20% memory outperforms TopK with 50% memory).
>
> ## What would happen with larger N
>
> We have added new experiments that set N to 50, 500, 1000, 10000 and reproduced the state frequency figure (Fig 3 A3) for these N. The new figures are in Appendix F.3, in the updated paper. But generally:
>
> - With N = 50, we cannot observe the long-tail behavior and state frequencies are approximately similar.
>     - With this level of capacity, a state would encode many different words.
>     - Specifically, the ratio of stopwords v.s. non-stopwords for each state does not vary much.
> - With N = 500, we start to observe long-tail behavior.
>     - This is to say, we can only observe the long-tail behavior when N is large enough.
>     - In the figures the differences between the tail and the head states become clear. "Dedicated" states emerge: there exist states that only encode stopwords or non-stopwords.
>     - The number of head states is approximately 150
> - With N = 1000 and 10000, we consistently observe long-tail behavior.
>     - The number of head states for N=1000 is approximately 250, and approximately 2000 for 10000 states.
>     - The ratio of head states states is approximately within range 20%-30% (150 for 500 states, 250 for 1000 states, 500 for 2000 states, 2000 for 10000 states)
>     - We also show (detailed below) that the head state ratio is related to the ratio of K1 (topK sum size) v.s. K2 (sample size) in our estimator.
> - Furthermore, we would like to point out that intuitively, larger N means states with finer-grained meanings and smaller levels of information compression. Smaller N means more level of information compression. We demonstrate this with a case study:
>     - With 1000 states, we observe 19 states that contain the subword unit "##es", which is associated with plural for nouns or third person singular for verbs. We observe the following state with corresponding word frequencies:
>         - jews 329 | children 271 | guns 239 | armenians 224 | cars 175 | christians 155 | drives 146 | lines 138 | keys 132 | turks 131 | disks 129
>     - This state splits into three different states when we set N = 10000, as is shown in the following:
>         - jews 441 | armenians 345 | arabs 31 | christians 27 | palestinians 19 | muslims 19 | israelis 15  —  this is an ethic group cluster, corresponding to the first word "jews" in the above cluster
>         - children 466 | kids 126 | parents 77 | child 19 | babies 18 | boys 13 | infants 12 | adults 9 | son 9 | families 8   —  this is an family word cluster, corresponding to the second word "children" in the above cluster
>         - guns 174 | tools 86 | weapons 80 | grenades 28 | knives 21 | skills 13 | boots 12 | axes 11 | bullets 10 —  this is an weapon word cluster, corresponding to the thrid word "guns" in the above cluster
>     - Intuitively, as N decreases, states merge with each other. Interestingly, it seems that tail states tend to merge with tail states, and head states tend to merge with head states, leading to a rather consistent ratio of the number of head v.s. tail states (about 20-30% percent as discussed above, we discuss this ratio below).

---

### Official Review · Reviewer_61KB · 2021-11-02

**Correctness:** 4
**Technical Novelty And Significance:** 2
**Empirical Novelty And Significance:** 2
**Recommendation:** 5
**Confidence:** 3

**Details Of Ethics Concerns:**

Paper utilizes large language models, which are known to have potential biases, but the analysis is not ethically concerning.

**Main Review:**

Strengths:

- The proposed Randomized Dynamic Programming approach provides a simple, well-founded approximation to the log partition function, as evidenced by Figure 2, especially in the case of dense distributions.
- Empirical analysis of State-Word connections [Figure 3] show meaningful linguistic information on semantic and syntactic roles; analysis of State-State Relations [Figure 4] provides information about language compositions such as infinitives and prepositional phrases.
- Generalizations to different graph structures are discussed in the Appendix, I believe the proposed approach could be extended to tree graphs straightforwardly
- As expected, memory and runtime for RDP from Appendix section E.4 are low (equivalent to TopK which sets $K_1 = K, K_2=0$)
- The proposed approach provides insights into linguistic roles and compositions without supervision unlike many preceding works which utilize supervised probing

Weaknesses:

- To the best of my knowledge, the RDP mechanism detailed in this paper combines two well studied approaches (i) importance sampling (i)TopK budget [Sun et al. 2019]. As a result, the methodological novelty of the sampling scheme is somewhat limited.
- Additionally, the latent network modeling approach is quite similar to standard well-studied approaches (linear-chain CRF, long-tailed prior on proposals, amortized variational inference) and provides limited novelty.
- As discussed in the paper, most linguistic information seems to be captured by 500 states. It would be helpful to compare experimental results for state-word and state-state relations among the Full, TopK, and RDP approaches which should all be feasible for 500 states. In particular, as seen in Figure 2, the benefits of RDP seem especially beneficial in Dense distributions. How does this translate to differences in the learned state-word and state-state connections?
- Marginal performance benefits in terms of Test PPL on MSCOCO [Table 1] and on Paraphrase generation on MSCOCO [Table 2] when compared to TopK and Full (for small # of states).



**Summary Of The Paper:**

This paper proposes a latent network analysis, using a proposed a Randomized Dynamic Programming (RDP) approach, to study the topology of the representation space of contextual embedding models. By constructing an unbiased estimator for the partition function with subsampled DP paths, the authors are able to reduce the memory complexity of forward-backward computation on Linear CRFs by 2 orders of magnitude. Using this RDP formulation, inference with structured latent variable models is scalable to 1000s of latent states. Latent states are modeled in a linear chain CRF with a long-tailed prior on transitions and learned using amortized variational inference similar to Fu et al. 2020, Li and Rush 2020. State-word connections in the representation space provide linguistic information that encapsulates syntactic and semantic roles and state-state connections correspond to the construction of phrases. The authors find anchoring states in the representation space that provide enough information to perform unsupervised paraphrase generation.

**Summary Of The Review:**

Interesting approach for inferring large-scale structured latent variable models in contextualized representation space. Empirical results are insightful but methodological contributions seem a bit limited in the context of prior literature.

---

> ### Author Response · Authors · 2021-11-15
> **Response to Reviewer 3 (61KB) (2/2)**
>
> # Comparison to TopK in 500 States, and benefits of RDP for training.
>
> - Firstly, training the LVMs with 500 states with exact forward is still not feasible (in our settings with 16G GPU we are only able to exactly compute 100 states). So we added a comparison to topK summation with 500 states in Appendix F.6. The results show that:
>     - TopK summation actually leads to posterior collapse where certain states become inactive. This phenomenon becomes more severe as the number of states increases (with 500 states, 10+ becomes inactive, with 2000 states, 600+ becomes inactive).
> - It is also important to differentiate the benefit of RDP for (a) partition estimation and (b) training LVMs. As is discussed above, when training LVMs, the sampling part of RDP should be interpreted as the exploration of the latent space (not just importance sampling for variance reduction), which encourages the model to increase the importance of tail states and makes them more frequent (see updated Appendix F.4 for more about exploitation-exploration tradeoffs.)
>
> # Other concerns about PPL and BLEU performance
>
> - When training LVMs, there are multiple objectives to tradeoff: (a) Partition function estimation, (b) ELBO, (c) PPL, (d) BLEU, (e) meaningful latent space, see the response to reviewer 2 for detailed discussion, also Alemi et. al. 2017 for how to tradeoff these objectives.
> - Basically, the common approach in the literature is to target a subset of these objectives (in our case, partition function and meaningful latent space), and try to balance objectives. So our experiments are more like a sanity check that shows our method is able to infer meaningful latent spaces without compromising PPL and BLEU (Also note that related work (Kim et. al. 2019) gives up on PPL for more meaningful latent structures).
>
> # References
>
> John Hewitt and Percy Liang. Designing and interpreting probes with control tasks. EMNLP 2019
>
> Alexander A. Alemi, Ben Poole, Ian Fischer, Joshua V. Dillon, Rif A. Saurous, Kevin Murphy. Fixing a Broken ELBO. ICML 2017
>
> Yoon Kim, Chris Dyer, Alexander Rush. Compound Probabilistic Context-Free Grammars for Grammar Induction. ACL 2019

---

> ### Author Response · Authors · 2021-11-15
> **Response to Reviewer 3 (61KB) (1/2)**
>
> We thank the reviewer for the detailed analysis. Here are our responses:
>
> # Clarification of our algorithmic contribution
>
> - It is important to clarify that our core contribution is not the sampling itself, but the design of the RDP algorithm, although sampling is an important ingredient thereof. The advantages of RDP can be summarized as follows:
>     - It is designed to be compatible with automatic differentiation and deep structured prediction libraries, which is discussed in Appendix B.
>         - In the discussion, we carefully rule out alternative methods and come to a conclusion that, to our knowledge, RDP is the only existing algorithm that is compatible with automatic differentiation and existing deep structured prediction libraries.
>     - Proved unbiasedness (Appendix A.1) and detailed variance analysis (updated in appendix A.2).
>         - In the updated appendix A.2, we theoretically decompose the variance into a tail term (which can be reduced with Rao-Blackwellization) and an error term (which can be reduced by a more correlated proposal).
>         - It is important to notice that, Rao-Blackwellization and Importance Sampling are just two basic tools for any estimators, it is the design of new estimators, rather than the basic tools used by new estimators, that makes our contribution.
>     - Although we focus on linear-chain CRFs, RDP is general and extends to any graph structures and any DP-based inference operations.
>         - To further validate this, we add experiments on: (1). new graph structure - partition estimation for tree-structured hypergraph; and (2). new inference operation - entropy estimation (which calls the forward as a subroutine).
>         - The extended algorithms are detailed in the updated Appendix D.1 and D.2,  results are added in the updated Appendix F.6 and F.7 where we consistently show that RDP is a more accurate estimator than the baseline topK with less memory consumption.
>     - Designed proposal based on the Zipfian nature of language (equation 10).
>         - We have further added studies about different proposals in the updated Appendix F.2, which show our designed proposal outperforms baseline proposals (including the uniform baseline).
>     - New exploration-exploitation interpretation when combining RDP with reparameterization (Page 5 the paragraph above the title of section 4)
>         - Increasing K1 would encourage the exploitation of confident states and increasing K2 would encourage the exploration of less confident states.
>         - We have further added experiments in Appendix F.4 that show our sampling strategy overcomes the posterior collapse problem induced by the baseline topK summation and encourages more active tail states (also see the response to Reviewer 4).
> - In summary, the main content of the paper may look superficially simple, primarily because we intended to make it accessible. In doing so we might have unintentionally played down the methodological novelty of our method. Yet It should be noted that all the above-discussed designs are not well-studied in the literature.
>
> # Clarification of our analytical contribution
>
> - It is important to note that our core contribution is not the modeling but the detailed analysis of the inferred structures. Specifically, under the context of Bertology, we note that:
>     - Our analysis is fully unsupervised and geometry-oriented, which:
>         - eschews the criticism (Hewitt and Liang 2019) of much previous work on supervised probes as they face the intrinsic dilemma about whether the discovered properties are intrinsic to the representation manifold or imposed by external supervision (see, page 2, last paragraph of Section 1, and Page 4 Section 4).
>         - discovers the intrinsic geometric structure of the pretrained LMs, rather than reinforcing our knowledge about well-known linguistic properties. This consequently leads to the following new observations:
>             - Morphology based clusters (Figure 3C)
>             - Clusters that encode tense (Figure 3C)
>             - Network edges showing phrasal constructions (Figure 4B)
>             - Long-tailed state behaviors (Figure 3A)
>             - The visualization of the latent network (Figure 4A)
> - To summarize, we provide new insight on fine-grained topological and linguistic properties of the representation space of pretrained language models. Our method can be used for visualizing, inspecting, and understanding contextualized representations.

---

> ### Comment · Reviewer_61KB · 2021-11-29
> **Response to rebuttal**
>
> I appreciate the response from the the authors, however my score recommendation remains the same. As other reviewers have mentioned, the contributions while interesting are not presented clearly to show novelty and significance. I would recommend that the authors make structural changes to the paper to emphasize the implementation advantages such as automatic differentiation with common libraries. Additionally, it may be more meaningful to look into NLP settings which have been traditionally modeled with graph/tree structures such as syntactic parsing tasks.

---

### Official Review · Reviewer_cUAQ · 2021-11-03

**Correctness:** 2
**Technical Novelty And Significance:** 3
**Empirical Novelty And Significance:** 2
**Recommendation:** 5
**Confidence:** 4

**Main Review:**

Strengths:
1. The paper is well written and does a good job connecting the method with prior work while also explaining the phenomenon that initially inspired the development of their method (the bias of top-k).
2. The potential connection to randomized automatic differentiation is both intriguing and I think will be useful in many applications. This paper does a good job of highlighting the limitations of our current tools and the authors did a good job documenting how these appear in the implementation.
3. The visualizations in this paper have strong execution and do a good job providing the reader with intuition regarding the structure of the discrete latent space the algorithm produces.

Weaknesses:
1. The main reason for approximating partition calculation is to scale LVMs to more states, but from experiment results, it doesn't seem that scaling LVMs to more states helps performance. For example, in table 1, the PPL of RDP-2k is only 0.19 better than RDP-100 which doesn't seem very significant, and in table 2, RDP-2k gets worse BLEU and iBLEU than RDP-50. Based on existing results in the paper, I cannot tell if it's because the approximation is bad, or because LVMs with more states do not get better performance, but either case is evidence against this work (in the former case, the proposed approximation is bad even though it's better than top-k; in the latter case, why would we want to scale LVMs then?). Another concerning result is that RDP-100 outperforms FULL-100 in table 1, and RDP-50 outperforms FULL-50 in table 2 other than self-BLEU. How could an approximation be better than exact inference?
2. The paper only focuses on a single application of the proposed randomized dynamic programming technique. When introducing such a broad technique, it seems natural to explore a range of applications to showcase the generality. It is unknown how it will perform when applied to other dynamic programming problems found in natural language processing or in machine learning.
3. I don't think the application of analyzing the representation space of pretrained language models makes sense: the CRF is only used to parameterize the inference network, but the generative model is trained on raw sentences (x) and does not use representations from the pretrained model. This setting contradicts the claim that "the discovered latent network is intrinsic to the representation manifold" (section 1, last paragraph), since the generative model is trained on x instead of representations r (Eq. 12). Even with a BERT with random weights, I think it is possible to learn meaningful clusters since the generative model is trained on x (so the posterior can still reveal meaningful structures). To me, a more natural setting would be to use an HMM to parameterize the generative model and perform exact marginalization over latent states using the proposed approach, which does not directly use raw words x but only uses representations r and would be truly "intrinsic to the representation manifold".
4. The paraphrasing experiment is not very clear. Where is bag-of-words used? What is the generative process?
5. Another issue with the paper is the lack of exploration of different sampling strategies. This is another place where I would be interested in results from a different task or domain. A stronger analysis of how the selection of K1 and K2 changes the result might also give insight into improving the method.

Minor issues:
1. It would be nice if you can compare to existing unsupervised clustering works such as Brown clustering which has a similar probabilistic formulation (https://github.com/percyliang/brown-cluster).
2. While the algorithm is inspired by randomized automatic differentiation, a method that provides an unbiased approximation of gradients, this method uses biased gradients of an unbiased approximation of the partition function?
3. Page 5, For experiment (2), I assume you meant (3)?

Questions for authors:
1. "we extract CRFs on-the-fly from different LVM training stages." Would the network learn log potentials to adapt to the approximate partition calculations so evaluation in this way becomes preferable to your approach?
2. Table 1 PPL, I assume you meant exp -ELBO for your models since true PPL is intractable under the generative model?  Or did you estimate it?
3. Fig. 2, how are the density curves drawn? Did you run your algorithm multiple times?

**Summary Of The Paper:**

This paper proposes a new way to greatly reduce the memory requirements of inference on discrete latent variable models (LVMs). It does this by estimating the key inference steps in dynamic programming through enumerating a small subset of nodes and using importance sampling for the rest of the nodes. This technique is generally applicable to many dynamic programming algorithms and enables scaling LVMs to much more states by reducing memory requirements. Empirically, this work shows that the proposed approach gets better partition estimates than existing methods using top-k approximations. This work also demonstrates an application of the proposed approach on discovering meaningful latent codes and latent trajectories from pretrained language models.

**Summary Of The Review:**

This paper is well-organized, has a novel contribution and I believe it will be of interest to various subgroups in the ICLR community. However, the experiment setting does not make much sense to me, and the empirical results are not convincing. I understand that I might be asking for too much for a single paper, but I think this paper would be much stronger with better experimental settings and more thorough analyses. I'd recommend the authors to split this paper into two papers. For example, the first paper can be an in-depth analysis of the proposed randomized dynamic programming algorithm with applications in different domains and different sampling strategies, and the second paper can be about structure discovery in pretrained models with a correct setup. In its current form, I am leaning towards rejecting this paper due to the issues mentioned before.

---

> ### Author Response · Authors · 2021-11-15
> **Response to Reviewer 2 (cUAQ) (5/5)**
>
> # Other important concerns
>
> Clarification on paraphrasing
>
> - The paraphrasing setting is basically the same as Fu. et. al. 2020 (except the encoder). Basically, the bag of words are used for:
>     - Initializing the decoder state, by setting the initial state to the average embedding of BOW.
>     - The decoder also attend to and copy from the BOW at each step
> - The generative process is just adding one conditional signal to each step of the model:
>     - p(x_t | z_t, ·) → p(x_t | z_t, BOW, ·), where · omits all previously generated x_t and z_t
>
> Brown cluster
>
> - Due to time limits, we have to make this a future work. Also, note that Brown clustering works on discrete word index so it would be a problem how to tailor Brown clustering for continuous embedding spaces ...
>
> "this method uses biased gradients of an unbiased approximation of the partition function?"
>
> - Yes
>
> "Page 5, For experiment (2), I assume you meant (3)?"
>
> - Yes, and we have updated that.
>
> "we extract CRFs on-the-fly from different LVM training stages." Would the network learn log potentials to adapt to the approximate partition calculations so evaluation in this way becomes preferable to your approach?
>
> - Not necessarily. The gradient of the log potential eventually comes from beta ELBO, rather than the approximated partition. Also, note that our additional studies on hypertrees and entropy estimation are simulated (no training here), and our approach still consistently outperforms the baseline topK summation method.
>
> "Table 1 PPL, I assume you meant exp -ELBO for your models since true PPL is intractable under the generative model? Or did you estimate it?"
>
> - We estimated PPL, not ELBO. PPL here is estimated with importance sampling (see Dyer et. al. 2016 for details) with 500 samples from the inference network (as the proposal for importance sampling). This is a standard practice in training LVMs (because ELBO in a sense is just a surrogate).
>
> "Fig. 2, how are the density curves drawn? Did you run your algorithm multiple times?"
>
> - Yes, we run the estimator 100 times with different samples from the proposal. Then density is from the KDE function from the Seaborn visualization library ([https://seaborn.pydata.org/](https://seaborn.pydata.org/)).
>
> # References
>
> Alexander A. Alemi, Ben Poole, Ian Fischer, Joshua V. Dillon, Rif A. Saurous, Kevin Murphy. Fixing a Broken ELBO. ICML 2017
>
> Yoon Kim, Chris Dyer, Alexander Rush. Compound Probabilistic Context-Free Grammars for Grammar Induction. ACL 2019
>
> Mark Steedman. The Syntactic Process. MIT Press. 2000.
>
> Songlin Yang, Yanpeng Zhao, Kewei Tu. PCFGs Can Do Better: Inducing Probabilistic Context-Free Grammars with Many Symbols. NAACL 2021
>
> Justin T. Chiu, Alexander M. Rush. Scaling Hidden Markov Language Models. EMNLP 2020
>
> Chris Dyer, Adhiguna Kuncoro, Miguel Ballesteros, Noah A. Smith. Recurrent Neural Network Grammars. NAACL 2016

---

> ### Author Response · Authors · 2021-11-15
> **Response to Reviewer 2 (cUAQ) (4/5)**
>
> # Additional study about different sampling strategies
>
> | Proposal | Dense | Intermediate | Long-tailed |
> | ---------- | ------- | --------------- | ------------- |
> | Uniform  | 0.6558 | 11.05 | 0.16 |
> | Local only | 2.9080 | 9.1711 | 0.4817 |
> | Global only | 0.4531 | 0.0961 | 0.1003 |
> | Local + Global | 0.0284 | 0.0173 | 0.0222 |
>
> - We now report experiments on how different sampling strategies interact with partition function estimation (same underlying CRFs as figure 2 in the main paper). Our local + global proposal (equation 10) outperforms other strategies in all unit cases. This table is also included in Appendix F.2.
>
> # Additional study about changing the K1 v.s. K2 ratio
>
> | K1 v.s. K2 | Dense | Intermediate | Long-tailed |
> | ----------- | ------- | ---------------- | ------------ |
> | 199:1     | 0.0701 | 0.0358 | 0.0468 |
> | 150 : 50 | 0.2729 | 0.0886 | 0.0330 |
> | 100 : 100 | 0.1458 | 0.1563 | 0.0691 |
> | 50 : 150 | 0.1284 | 0.0889 | 0.0804 |
> | 1 : 199 | 0.1930 | 43.43 | 5.4479 |
>
> - We now examine how the K1 v.s. K2 ratio interacts with the partition function estimation (same underlying CRFs as figure 2 in the main paper). Generally, this experiment seems inconclusive.
> - Theoretically, it would be better if K1 is equal to the number of modes (peaks) of the underlying distribution. Yet it is inherently hard to know the exact number of modes.
> - A general suggestion for parameter tuning would be to start with a half-half split and use it as a baseline.

---

> ### Author Response · Authors · 2021-11-15
> **Response to Reviewer 2 (cUAQ) (3/5)**
>
> # Clarifications on the generative model
>
> - "the generative model ... does not use representations from the pretrained model"
>     - This is a misunderstanding. The generative model shares the embedding matrix of the latent states.
>     - This is to say, if the state embedding is meaningful enough, i.e., it contains enough information for the reconstruction of its corresponding word, then the decoder would give a higher conditional likelihood of the word, i.e., p(x_t | z_t). If the state embedding is not informative, then the decoder would simply ignore the latent state, and collapse to an unconditional language model.
> - "Even with a BERT with random weights, I think it is possible to learn meaningful clusters"
>     - This is also a misunderstanding. We have added a comparison of the reconstruction likelihood, i.e., p(x_t | z_t), between a pretrained BERT and a random BERT. The training curve is shown in Appendix F.5. Generally, the reconstruction likelihood with a pretrained BERT is clearly higher than a random BERT, which means that the states are more informative than states from random Bert. After convergence, we have the reconstruction NLL of pretrained BERT = 1.73 v.s. random BERT = 2.58 (smaller is better)
>     - Manual inspection also shows that although there is a certain level of clustering happens with a randomly initialized BERT (because of the strong approximating power of neural networks),  these clusters are much noisier than pretrained BERT. As an example:
>         - with a random BERT we find out this plural cluster with words:
>             - people 1363 | lines 1361 | things 611 | children 452 | writes 409 | files 400 | systems 389 | jews 382 | programs 323
>         - with a pretrained BERT, words in the above clusters are better separated to the following dedicated clusters:
>             - turks 222 | armenians 206 | jews 186 | keys 171 | muslims 151 | arabs 123 | christians 93 (proper names)
>             - opinions 340 | problems 154 | rules 140 | views 113 | things 104 | laws 99 | events 97 (abstract concepts)
>             - others 13 | us 9 | children 9 | jews 8 | years 8 | groups 7 | pages 7 | humans 7 (nouns)
> - We have updated the full list of state-word relations in the supplementary materials. We encourage the reviewer to browse through.
> - Additionally, it is not clear how to build an HMM with a BERT encoder — we are not aware if there exists such work, but to our knowledge, common practice is to build a discriminative model (linear-chain CRFs and treeCRFs) upon a pretrained encoder. However, for generative models (HMMs and PCFGs), their neural parameterization is intrinsically incompatible with pretrained encoders, and existing work does not use BERT as an encoder (Chiu and Rush 2020, Kim et. al. 2019, Yang et. al. 2021, ).

---

> ### Author Response · Authors · 2021-11-15
> **Response to Reviewer 2 (cUAQ) (2/5)**
>
> # Additional evaluation on extensions to tree-structured hypergraphs and entropy estimation
>
> We agree with the reviewer that more evaluation should be done on more graphs and inference operations. So we added experiments on:
>
> - Tree-structured hypergraphs, which cover dependency trees and PCFGs and estimate the partition function (equations in the updated Appendix D.1, results in Appendix F.7)
> - Entropy estimation on chain structures, which calls the randomized forward as a subroutine, and estimates the entropy (equations in the updated Appendix D.2, results in Appendix F.6).
>
> Together with the existing randomized forward, we cover:
>
> - Two most common structures in NLP: chains and trees
> - Two common inference operations: partition function and entropy
> - Furthermore, these techniques are directly compatible to reperameterization, which is yet another common inference operation, and used for training LVMs in our paper.
>
> We hope this gives enough evidence for the scope of our method.
>
> ## Results on entropy estimation with randomized entropy DP computation
>
> Settings:
>
> - We re-use the three linear-chain CRFs from our paper Figure 2 for consistency.
> - We implement a randomized entropy algorithm and compare it to its topK counterpart.
> - We draw density figures visualizing bias and variance in the updated Appendix F.7 (similar to Figure 2 in the main paper).
> - We also report the mean square error (MSE) of different estimators (recall that MSE is the combination of bias and variance) in appendix F.7. The MSE table is copied here:
>
> | Methods | Dense | Intermediate | Long-tail |
> | ---------- | --------- | ------------- | --------   |
> | Entropy TopK 20% Mem | 443.7 | 84.35 | 8.0115 |
> | Entropy TopK 50% Mem | 131.8 | 22.1 | 1.8162 |
> | Entropy RDP 1% Mem | 5.9256 | 1.9895 | 0.6914 |
> | Entropy RDP 10% Mem | 2.1168 | 1.2989 | 0.3167 |
> | Entropy RDP 20% Mem | 1.3267 | 0.7305 | 0.2071 |
> | Entropy RDP 50% Mem | 0.3017 | 0.1461 | 0.0631 |
>
> - From the table we see that our randomized entropy DP consistently outperforms the topK summation baseline with less memory (ours with 1% memory outperforms the baseline with 50% memory)
>
> ## Results of randomized inside for partition function estimation of tree-structured hypergraphs
>
> Settings:
>
> - We simulate three unit cases of dependency tree CRFs (dense, intermediate and long-tail) by controlling the entropy.
> - We set the number of latent states to 2000.
> - We use a uniform proposal for simplicity, and set K1 (top K sum size) = K2 (sample size).
> - We use RDP with 20, 200, 400 states, which correspond to 1%, 10%, and 20% of the full number of states.
> - We use topK summation as our baseline with 500 and 1000 states (25% and 50% of the full number of states).
> - We draw density figures visualizing bias and variance in the updated Appendix F.6 (similar to Figure 2 in the main paper).
> - We also report the MSE of different estimators in Appendix F.6. The MSE table is copied here:
>
> | Methods | Dense | Intermediate | Long-tail |
> | ---------- | --------- | ------------- | --------   |
> | Inside TopK 20% Mem | 36.1275 | 27.4351 | 21.7788 |
> | Inside TopK 50% Mem | 2.8422 | 2.4043 | 2.0479 |
> | Inside RDP 1% Mem | 26.3312 | 37.6698 | 48.8638 |
> | Inside RDP 10% Mem | 1.1937 | 1.5307 | 1.3843 |
> | Inside RDP 20% Mem | 0.4455 | 0.5449 | 0.5997 |
>
> - From the table we see that our randomized inside consistently outperforms the topK summation baseline with less memory (ours with 10% memory outperforms the baseline with 50% memory)

---

> ### Author Response · Authors · 2021-11-15
> **Response to Reviewer 2 (cUAQ) (1/5)**
>
> # Clarification on performance metrics and objectives of learning LVMs
>
> - To answer the first concern about the experimental results, it is important to clarify the following objectives involved in training latent variable models:
>     1. The partition function of the inference model, which is the goal of our RDP
>     2. ELBO training objective, which calls the partition as a subroutine.
>     3. NLL testing objective, which is the major objective of density estimation, but not necessarily the major objective of LVMs, as many LVMs target a meaningful latent space.
>     4. BLEU objective, which is the major objective of text generation.
>     5. Meaningful latent space, which is a major interest of LVMs, and is more or less why LVMs even exist in the first place.
> - In our paper, our major objective is (1) an efficient estimation of the partition function, and (5) a meaningful latent space. In this sense, NLL and BLEU objectives function more like a sanity check to demonstrate that our method is indeed effective.
> - These objectives correlate and interact with each other in complicated ways (which on its own is an active research area, see Alemi et. al. 2017 for a detailed explanation), so targeting one or a subset may result in undesirable tradeoffs.
>     - For example, in Kim et. al. 2019, the authors target meaningful latent trees *at the cost of NLL*. Yet this is indeed acceptable because the target is the latent space.
>     - Similarly, we aim to obtain meaningful latent spaces. Our methods induce meaningful latent networks without compromising NLL.
> - In our work, the interaction of these objectives, and the comparison to baselines can be interpreted as:
>     - RDP gives a better partition estimation than the baseline topK summation.
>     - This leads to better PPL (or equally NLL) than models trained with topK summation. Note that although the margin is not large, it is consistent.
>     - RDP's better NLL than exact forward should not be that surprising: randomization induces noises in partition estimation (which is an intermediate step for training LVMs), but not necessarily influence the downstream NLL in a negative way. In fact, there are plenty of examples in deep learning where intermediate step noise-injection improves downstream performance. The most typical example might be dropout, as it replaces exact computation with masked values. Ours can be viewed as dropping out states that are not sampled.
>     - Increased size of latent space does not necessarily lead to a better NLL (and similarly BLEU). This should be acceptable in our case because our goal is to scale the size of the latent states (not increasing NLL) and induce meaningful latent spaces. In fact, it would not make sense to use LVMs for better PPL as there are already way better models for PPL like GPT2.
> - We want to scale LVMs because we want meaningful latent spaces, and there is a lot of interesting linguistic phenomena which emerge only when the state space is large enough. From an information compression point of view, as the number of states increases, we induce states with different linguistic granularities:
>     - When the latent states are 20, then these states are approximately the same size as POS tags; if 200, then approximately the size of fine-grained named entity recognition and semantic role labeling.
>     - When we scale the states to 2000, these states are approximately supertags (Steedman 2000), or meta-words that encode finer meaning.
>     - For human inspection and interpretability, we need fine-grained supertags since they give more specific linguistic roles. If the states are not large enough, words with different linguistic roles would merge into the same state, making states hard to interpret. This is why 2000 is an appropriate level of granularity since it gives *detailed enough information compression*.
> - We further highlight some of the new insights we gain from scaling the number of states to 2000 which have not been previously observed in the literature, which further demonstrate the effectiveness of learning meaningful latent spaces:
>     - Morphology based clusters (Figure 3C)
>     - Clusters that encode tense (Figure 3C)
>     - Network edges showing phrasal constructions (Figure 4B)
>     - Long-tailed state behaviors (Figure 3A). Note these emerge only when N, the number of states, is large enough (see our response to Reviewer 4 about controlling N).
>     - The visualization of the latent network (Figure 4A)

---

### Official Review · Reviewer_irMJ · 2021-11-05

**Correctness:** 3
**Technical Novelty And Significance:** 3
**Empirical Novelty And Significance:** 2
**Recommendation:** 3
**Confidence:** 4

**Details Of Ethics Concerns:**

No concerns.


**Main Review:**

To put it simply, BERT or GPT-2 like models learn continuous contextual representations of words, so sentences. In this paper, linear-chain Conditional Random Field (CRF) model is considered with a number of hidden states which is relatively small w.r.t. number of words, with each hidden state representing a set of words. Just like word indices are discrete, one can think of the indices of the states of CRF as discrete even though the hidden state vectors are high dimensional continuous vectors, hence the authors project this approach as of  discrete latent structures. Essentially, all this approach is doing is clustering words based upon their contextual embeddings, and then building a graph of transition between those states. In the experiments, the number of states is 2000 while the number of words in BERT is 32k, so there isn’t a significant reduction in the number of states if one were to consider each word as a state. There isn’t any novel contribution in this idea, nor a value of such analysis, at least from the perspective to advancing science. Even empirically, I don’t see any interesting insights from the analysis.

Further, since CRF models are trained using dynamic programming which is efficient in itself but it has quadratic cost in the number of states. It is proposed to sub-sample upon some of the sequences with low weights, while retraining the ones with high weights. This approach is just a heuristic, with many possible flaws. For instance, it is proposed to compute the sequences with high weights absolutely without recursive formulation, which should be expensive. Sampling strategy is not explained well, and I believe challenging to obtain for NLP like high dimensional problems. Besides, the approximation is applied at coarser level, not accounting for tree like structure emerging from recursion, big sequences as child nodes and subsequences as parent nodes. Intuitively, something like brand and bound could have helped if there was sampling accounting for tree structure. It is also worth noting that the proposed sampling approach is not novel but explored in related problems. Theoretical analysis on zero bias is not enough, and there isn’t sufficient analysis for ensuring low variance.

**Summary Of The Paper:**

Contextual embeddings of words are mapped to a sort of cluster of word embeddings with each one corresponding to one possible state in a linear CRF model. The idea is to build a network between the cluster of words (states of CRF) for analysis the manifold of language. To reduce quadratic cost of CRF, a sampling based approach is proposed to compute the partition function.

Empirical analysis is not very insightful, and the contributions are incremental.


**Summary Of The Review:**

Probably, this is a preliminary work, and it could become a good paper with lot more research efforts put into it.

Algorithmic contribution of sampling is interesting but incremental and not throughly studied. Empirical analysis is not very insightful either. I don’t see much justification in mapping 32k contextual embeddings to 2k latent embeddings to build a so called latent representation.

---

> ### Author Response · Authors · 2021-11-15
> **Response to Reviewer 1 (irMJ) (4/4)**
>
> ## Our analytical contributions, in the context of Bertology literature
>
> - There are abundant papers analyzing the representations of pretrained language models (see Rogers et. al. 2020 for a survey on 150+ papers). A paper analyzing the syntactic subspaces of BERT even won a best short paper award in NAACL 19 (Hewitt et. al. 19).
> - It is important to understand and inspect the internal structure of blackbox PLMs. It is the opaqueness of these models that has caused significant concerns with regard to interpretability, deployment, and ethics.
> - There has been increased interest in unsupervised methods and the geometric properties of BERT-like spaces in the field of analyzing pretrained language models.
>     - The increasing interest in unsupervised methods is because supervised probes inevitably face the dilemma about whether the discovered properties are intrinsic to the representation manifold or imposed by external supervision. Our method is fully unsupervised, this eschewing the criticism (Hewitt and Liang 2019) of much previous work on supervised probes.
>     - The increased interest in geometric properties is in contrast to conventional linguistic properties. This is because a lot of existing work only reinforces our knowledge about the well-known linguistic properties, rather than properties specific to the learned representations. Our work focuses on the intrinsic geometric structures, and we emphasize that many of our discovered states are NOT reinforcing conventional properties by supervised probes.
> - We further highlight some of the new insights that we gain from inspecting the topology of our  induced network (as a contrast, see Tenney et. al. 2019 to get a flavor of previous probing work):
>     - Morphology based clusters (Figure 3C)
>     - Clusters that encode tense (Figure 3C)
>     - Network edges showing phrasal constructions (Figure 4B)
>     - Long-tailed state behaviors (Figure 3A)
>     - The visualization of the latent network (Figure 4A)
> - To summarize, we provide new insights on the fine-grained topological and linguistic properties of the representation space of pretrained language models. Our method can be used for visualizing, inspecting, and understanding contextualized representations.
>
> ## We would really appreciate it if the reviewer could go over the paper again and re-evaluate it. We are willing to provide any further clarifications and explanations requested.
>
> # References
>
> Justin T. Chiu and Alexander M. Rush. Scaling Hidden Markov Language Models. EMNLP 2020
>
> Zhiqing Sun, Zhuohan Li, Haoqing Wang, Di He, Zi Lin, and Zhihong Deng. Fast structured decoding for sequence models. NeurIPS 2019
>
> Songlin Yang, Yanpeng Zhao, Kewei Tu. PCFGs Can Do Better: Inducing Probabilistic Context-Free Grammars with Many Symbols. NAACL 2021
>
> John Hewitt, Christopher D. Manning. A Structural Probe for Finding Syntax in Word Representations. NAACL 2019
>
> Anna Rogers, Olga Kovaleva, Anna Rumshisky. A Primer in BERTology: What we know about how BERT works. TACL 2020
>
> John Hewitt and Percy Liang. Designing and interpreting probes with control tasks. EMNLP 2019
>
> Xingyu Cai, Jiaji Huang, Yuchen Bian, Kenneth Church. Isotropy in the Contextual Embedding Space: Clusters and Manifolds. ICLR 2021
>
> Mark Steedman. The Syntactic Process. MIT Press. 2000.
>
> Ian Tenney, Patrick Xia, Berlin Chen, Alex Wang, Adam Poliak, R Thomas McCoy, Najoung Kim, Benjamin Van Durme, Samuel R. Bowman, Dipanjan Das, Ellie Pavlick. What do you learn from context? Probing for sentence structure in contextualized word representations. ICLR 2019

---

> ### Author Response · Authors · 2021-11-15
> **Response to Reviewer 1 (irMJ) (3/4)**
>
> # Justification of our contribution and reiteration of background literature and linguistic basics
>
> ## Justification for mapping 32k contextual embeddings to 2k latent embeddings
>
> - To understand the meaning of 2K latent states, it is important to realize the hierarchical nature of human language. The goal is not just increasing or reducing states. The goal is to clarify the linguistic meaning of states at *different levels of granularity*, and we argue that 2K states are an appropriate level of abstraction for linguistic concepts. From an information bottleneck point of view, different granularities have different linguistic interpretations:
>     - When the latent states are 20, then these states are approximately the same size as POS tags which give a more coarse, high-level summary of word meaning. CRFs at this level are tractable on GPUs and previous work has studied this.
>     - When states are 200, they provide an intermediate level compression of lexical information, this is similar to the size of fine-grained named entity recognition and semantic role labeling.
>     - If the latent states are 2000, they behave like supertags (Steedman 2000), or meta-words that encodes finer meaning distinctions.
>     - Although the vocabulary size of BERT is 30K, its output space is effectively larger due to the influence of context. One word may have different linguistic roles and different words may share the same linguistic roles. *This is precisely what our induced networks capture*.
>     - For human inspection and interpretability, we need fine-grained supertags since they reveal specific linguistic roles. If the states are not large enough, words with different linguistic roles would merge into the same state, making states hard to interpret. This is why 2000 is an appropriate level of granularity since it gives *detailed enough information compression*.
>     - A study of controlling the number of states has also been added in Appendix F.3. Also see the discussion with reviewer 4.
>
> ## Our algorithmic contributions, in the context of the probabilistic inference literature
>
> - The scaling of discrete model inference has been the subject of previous work on multiple tasks. For example:
>     - Chiu and Rush. 2020. Their method is tailored for HMMs and requires pre-clustered states. Our method does not have the pre-clustering restriction and is applicable to any graph structure.
>     - Sun et. al. 2019, our baseline. The major problem with their method is that it is significantly biased while ours is not.
>     - Yang et. al. 2021. Their method is tailored for PCFGs and only scales to hundreds. Ours is applicable to any graph structure and can scale to tens of thousands.
> - In summary, our method provides a principled solution to scaling discrete model inference with the following advantages
>     - Compatibility with automatic differentiation
>     - Applicability to any graph structure and inference operations
>     - Scalability to tens of thousands of states
>     - Exploitation of the Zipfian property of human language.

---

> ### Author Response · Authors · 2021-11-15
> **Response to Reviewer 1 (irMJ) (2/4)**
>
> # Misunderstanding of the basics in this paper (cont.)
> - "This approach is just a heuristic, with many possible flaws. "
>     - Our approach is far more sophisticated than "just a heuristic". There are multiple design considerations in RDP, specifically:
>         - It is built upon well-established techniques, previously discussed in Kool et. al. 2020 and Sun et. al. 2019. RDP uses these techniques as a solid foundation.
>         - We have considered and ruled out multiple alternative attempts before building the RDP algorithm (detailed in appendix B), mostly due to the compatibility of automatic differentiation.
>         - RDP is carefully designed to:
>             - Be compatible with Automatic Differentiation (see more in Appendix B), which has been a major challenge in previous work and not thoroughly discussed previously.
>             - Extend to multiple graph structures, see Appendix D for extensions to general sum-product and Appendix F.6 for additional experiments on tree-structured hypergraph (PCFGs) partition estimation and Appendix F.7 for additional experiments results on entropy estimation. Our method consistently outperforms the TopK summation baseline in all the graph structures (chains and trees) and all inference operations (partition, reparameterization, entropy estimation).
>             - Scale to tens of thousands of states. This is further validated in the updated Appendix F.3 where we scale the number of states to 10K.
>             - Exploit the Zipfian property of language, which is discussed in the construction of the proposal (equation 10). The Zipfian property is also verified in experiments (Section 5.2, figure 3).
> - "It is also worth noting that the proposed sampling approach is not novel but explored in related problems"
>     - We do not claim that the sampling method in equation 2 is novel. Rao-Blackwellization and importance sampling are just basic tools and are used in multiple places. In our paper, we explicitly write "it is the underlying basis of many Monte Carlo estimators in various settings"
>     - It is the *design of new algorithms* (for example, in Kool et. al. 20, gradient estimation, in our case, RDP) that makes the core contribution, and equation 2 is just an element.
>     - There are multiple equally important elements (compatibility of Autodiff, extendability, .etc) combined together, jointly making our proposed RDP nontrivial.
> - "there isn’t sufficient analysis for ensuring low variance."
>     - In Figure 2, the variance of the estimates (range of the peaks) is smaller than the distance between the grey lines (baseline estimator) and the red line (target value). This is empirical evidence for low variance.
>     - The source of variance is discussed on page 2 the last paragraph, visualized in Figure 1 by green bars, and detailed in the updated Appendix A, theorem A.2 to A.4.
>     - For the estimation of the linear-chain partition function estimation, we now provide an updated mean square error comparison between RDP and TopK in the updated Appendix F.1 (recall that MSE is the combination of bias and variance). RDP with a 1% memory budget achieves less MSE than TopK with a 20% memory budget, specifically, in the dense case, RDP MSE v.s. TopK MSE = 0.14 v.s. 3.87, in the intermediate case, RDP v.s. TopK = 0.06 v.s. 1.01, in the long-tail case, RDP v.s. TopK = 0.07 v.s. 0.16
>     - We have further added experiments showing our method achieves lower MSE than the baseline TopK summation for:
>         - Tree-structured hypergraph partition estimation, Appendix F.6.
>         - Chain-structure entropy estimation, Appendix F.7

---

> ### Author Response · Authors · 2021-11-15
> **Response to Reviewer 1 (irMJ) (1/4)**
>
> ## Unfortunately, we cannot agree with this review because there seems to be some confusion with respect to our method and its motivation which we attempt to dispel below
>
> # Misunderstanding of the basics in this paper
>
> - "it is proposed to compute the sequences with high weights absolutely without recursive formulation"
>     - No, our method is recursive, and we have discussed its recursive nature in several places in our paper. Specifically:
>         - Page 3, the paragraph above equation 4, we write "Our key insight is to *recursively* use the memory-efficient randomization ..."
>         - Page 3, the paragraph above equation 6, we write "the key *recursion* of our Sampled Forward uses ..."
>         - Page 3, Figure 1B is titled "Sampled Forward *Recursion*"
>         - Page 3, equation 6 itself is a recurve computation.
>         - Appendix A.1, proof of unbiasedness, we use proof by induction, which is recursive by itself.
> - "It is proposed to sub-sample upon some of the sequences "
>     - We do not sample *sequences,* we sample *states* at each recursion step, and compute all possible sequence combinations through DP.
> - "the approximation is applied at coarser level, not accounting for tree like structure emerging from recursion"
>     - Our method accounts for *all possible tree combinations* over important states and corrects the bias of all states that are not sampled (thus associated trees) by dividing the sample probability.
>     - We have given a detailed analysis of the sources of errors on Page 2, last paragraph, Figure 1 B, page 3 last paragraph, and have updated a theoretical variance analysis in Appendix A.2.
> - "Sampling strategy is not explained well"
>     - A large portion of sections 2, 3, Figure 1, Appendix A and C explain the sampling strategy, specifically:
>         - The first half of Section 2, Speeding Summation by Randomization, is used to explain how to estimate tail summation by sampling. The sampling procedure is also visualized in Figure 1A.
>         - In the second half of Section 2, The Sampled Forward Algorithm, equations 4-7  explain how to use the simple sampling recursively within DP. This is also visualized in Figure 1B titled "Sampled Forward *Recursion* "
>         - Bias and variance are explained in Section 2 (lower part of page 2), also in Appendix A (we have also added theoretical analysis of variance per request in Appendix A.2).
>         - Proposal construction is discussed in equation 10. Note how we exploit our knowledge about language to construct this proposal.
>         - Step-by-step implementation guidance is provided in Appendix C.
> - "Sampling strategy is ... challenging to obtain for NLP like high dimensional problems"
>     - The *proposal for NLP* is specifically designed in equation 10. It exploits the Zipfian nature of language, and its effectiveness is verified in the experiments Figure 2.
>     - More comparison of our sampling strategy against three baselines is added in the updated appendix F.2. Our proposed proposal (equation 10) consistently outperforms the baseline uniform proposal in terms of MSE in all unit cases. Specifically, in the dense setting, our proposal MSE v.s. uniform proposal MSE = 0.02 v.s. 0.65, in the intermediate setting, ours v.s. uniform = 0.01 v.s. 11.05, on the long-tail setting, ours v.s. uniform = 0.02 v.s. 0.16

---

> ### Comment · Reviewer_irMJ · 2021-11-23
> **Post-rebuttal**
>
> I thank the authors for the detailed responses and the clarifications. Though, my overall recommendation remains same. The only interesting aspect in the paper is "randomized dynamic programming", and it has been pursued in the field. Overall the paper contributions are not exciting or impactful.

---

### Author Response · Authors · 2021-11-15
**Summary of Paper Updates**

Below is a list of paper updates (all added in the appendix) per reviewers' requests:

- Appendix F.1, Mean-square error table showing performance v.s. memory tradeoffs (Review 4)
- Appendix F.3, results on controlling the number of states (Review 3 and 4)
- Appendix F.4, controlling ratio of K1 (sum size) v.s. K2 (sample size) (Reviewer 1, 2 and 4)
- Appendix A.2, theoretical analysis of variance (Reviewer 1 and 4)
- Appendix D.1, D.2, F.6, F.7, extensions of RDP to more structures (hypertrees) and more inference operations (entropy) (Reviewer 1, 2 and 3)
- Appendix F.5, comparison to a randomly initialized BERT (Reviewer 2)
- Appendix F.2, comparison between different sampling strategies (Reviewer 1, 2 and 3)

---

### Decision · Program_Chairs · 2022-01-20

**Decision:**

Reject

**Comment:**

The paper introduces a technique for randomised dynamic programming and uses it to scale a latent variable model that enables interpreting the hidden states of large pre-trained models for text representation and generation.

The current version needs to be improved with regards to scope, which can be seen by the various confusions that it triggered, and which the authors tried to address in the rebuttal phase. It is somewhat unclear to all of us (myself included) whether the paper is about i) randomised dynamic programming (RDP), or ii) RDP's role in a particular LVM (with a CRF posterior approximation), or iii) RDP+LVM's ability to interpret deep Transformer models? Empirically, the paper is much more about (iii), somewhat about (ii, e.g. Table 1), very little about (i, e.g. Figure 2).

*Because the scope is now confusing*, the current version sometimes comes across as relatively incremental or even incomplete:

* Should the authors embrace interpretation. The overall strategy is *very interesting*, and it scales a neat model precisely in the way it needs to be scaled to do what it's meant to do, but this would change the focus of the paper, RDP would be all but a means to an end, and perhaps other techniques for interpretation would be needed.

* Should the authors embrace RDP itself (disentangled from its application to model interpretation). Some of us felt like the randomisation technique on its own is not too surprising (given the work of [Liu et al](http://proceedings.mlr.press/v97/liu19c/liu19c.pdf), for example), and, regardless of that, to push for RDP's significance, the paper would need more comparisons. The only alternative to RDP investigated in the paper is a heuristic top-K gradient. There are deterministic gradients that are less heuristic, and which may become unbiased eventually as training progresses, see for example [[1]](https://aclanthology.org/D18-1108/) and [[2]](https://papers.nips.cc/paper/2020/hash/887caadc3642e304ede659b734f79b00-Abstract.html).

In the first round of reviews there were some comments that questioned the paper's fitness to ICLR, I would like to remark that this has been clarified, and the paper targets a problem of clear relevance to the conference.

I would personally like to add a minor comment: it would be nice to acknowledge some older literature on randomised DPs (see for example [[3]](https://papers.nips.cc/paper/2009/hash/e515df0d202ae52fcebb14295743063b-Abstract.html) and [[4]](https://aclanthology.org/N10-1028/)).